# Soil Deposition of Atmospheric Hydrogen Constrained using Planetary Scale Observations

Alexander K. Tardito Chaudhri[1] and David S. Stevenson[1]

[1]The University of Edinburgh, King's Buildings, Alexander Crum Brown Rd, Edinburgh EH9 3FF

**Correspondence:** Alexander K. Tardito Chaudhri (achaudH2@ed.ac.uk)

**Abstract.** Quantifying soil deposition fluxes remains the greatest source of uncertainty in the atmospheric $H_2$ budget. A new method is presented to constrain $H_2$ deposition schemes in global models using observations of the zonal mean $H_2$ distribution and seasonality. A 'best-fit' scheme that reproduces the observed zonal-mean seasonality of atmospheric $H_2$ at the planetary scale is found by perturbing a prototype deposition scheme based on soil temperature and moisture dynamics. Comparing the best-fit and prototype schemes provides insight for how the prototype scheme may be improved to better reproduce observed seasonality. The $H_2$ signal driven by the prototype scheme is accurate compared to observations in the Northern Hemisphere extra-tropics but shows discrepancies in the Southern Hemisphere, with too high surface mixing ratios and too weak seasonality. A best-fit scheme indicates that the function capturing the soil microbial consumption of $H_2$ requires a shift of +2 to +3 months in the seasonality in the tropics, with peak uptake shifting from February to April in the Southern Tropics, and from August to October in the Northern Tropics, compared with a prototype scheme sensitive to seasonal soil moisture driven by the shifting of the ITCZ. New constraints on the $H_2$ surface flux at low-latitudes are key to accurately modelling the $H_2$ cycle in the Southern Hemisphere.

## 1 Introduction

In recent years there have been widespread announcements of new plans for hydrogen energy systems (Warwick et al., 2023). As such, future hydrogen emissions are expected to increase, in large part due to fugitive emissions from infrastructure (Ocko and Hamburg, 2022; Esquivel-Elizondo et al., 2023). Although the symmetrical $H_2$ molecule is not itself a greenhouse gas, its presence depletes atmospheric OH that would otherwise oxidise methane (Warwick et al., 2023). Hydrogen oxidation by OH additionally contributes to the formation of the greenhouse gases ozone and water, with the latter having a significant warming effect in the otherwise dry stratosphere (Sand et al., 2023; Warwick et al., 2023). However, despite the increasing number of modelling studies that have provided new insights into the atmospheric chemistry of $H_2$, there remain large uncertainties in evaluations of the Global Warming Potential (GWP) of $H_2$ (Sand et al., 2023; Derwent, 2023). Different recent estimates for the GWP over a 100 year time horizon include: Sand et al. (2023), $11.6 \pm 2.8$; Derwent (2023), 7.1-9.3; and Warwick et al.

(2023), $12 \pm 6$. Quantifying the soil sink, and counterbalancing $H_2$ emissions, have remained the main source of uncertainty in the atmospheric $H_2$ budget (cf. Novelli, 1999; Sand et al., 2023) and hence the lifetime of $H_2$, which generates large uncertainty in the GWP calculations that are relied on for understanding the climate impacts of $H_2$ applications.

Present-day (2010s) surface $H_2$ mixing ratios are c.550 ppb (Pétron et al., 2023), and showed an increasing trend of about 2.5 ppb yr$^{-1}$ in the latter part of the 20$^{th}$ century (Patterson et al., 2020). Multi-decadal increases in $H_2$ can mainly be attributed to increases in methane oxidation, while sub-decadal variations are more related to changes in the soil sink and $H_2$ emissions (Derwent et al., 2023; Paulot et al., 2024). $H_2$ concentrations show a distinct latitudinal variation, with values about 50 ppb higher in the Southern Hemisphere (SH) compared to Northern Hemisphere (NH) high-latitudes. They also show a characteristic seasonal cycle; outside the tropics $H_2$ generally peaks in the summer, with a monthly mean peak-to-peak magnitude of 30-60 ppb in the NH and 15-30 ppb in the SH.

To produce the same observed atmospheric $H_2$ concentrations, a stronger soil sink implies greater emissions and a shorter atmospheric lifetime (Hauglustaine et al., 2022; Ehhalt and Rohrer, 2013; Sand et al., 2023). In a comparison of six atmospheric chemistry models with imposed boundary layer $H_2$ concentrations, Sand et al. (2023) evaluated an uncertainty contribution to the GWP of 18% of the mean due to uncertainty in the soil sink.

While constraining the annual mean planetary soil sink constrains the lifetime of $H_2$ in the bulk atmosphere, the observational data contains additional useful information about the latitudinal distribution and seasonal variation of $H_2$, that we exploit here. We filter these observations to decompose the observed $H_2$ signal into a 2012-2018 mean background state and a seasonal cycle (Sec. 3).

Several different deposition schemes to model the soil sink of $H_2$ exist (e.g. Sanderson et al., 2003; Paulot et al., 2021; Ehhalt and Rohrer, 2013; Bertagni et al., 2021). These schemes are typically based on laboratory or field studies of a small number of deposition flux samples (e.g. Yonemura et al., 2000; Meredith et al., 2017) and model deposition velocities with functions of soil texture, water content, and temperature. There are also indications that soil carbon content is important in determining the $H_2$ deposition velocity (Khdhiri et al., 2015; Karbin et al., 2024). Here we provide a method to evaluate deposition schemes at the planetary scale. Observational data of surface $H_2$ measurements provides time-series of surface mixing ratios from a globally distributed set of sites (Pétron et al., 2023). We extend previous analysis of the seasonality of individual station measurements (Novelli, 1999) and on the combined roles of tropical biomass burning, deposition, and convective uplift in contributing seasonal variation (Hauglustaine and Ehhalt, 2002; Yashiro et al., 2011), through an analysis that accounts for the continuous variation of seasonality with latitude. This extends the work of Xiao et al. (2007) in spatial and temporal resolution, who have previously decomposed $H_2$ sources and sinks based on the seasonality in tropical and extra-tropical regions.

Our results and analysis also complement Paulot et al. (2024), who have revised $H_2$ emissions and deposition estimates to better-reproduce the $H_2$ distribution and seasonality of clustered observations. They identified major gaps in our understanding soil removal of $H_2$. We analyse the soil uptake arising from a model constrained to reproduce a latitude varying $H_2$ seasonality, while relaxing assumptions on the estimated soil uptake.

The annual-mean distribution and seasonality of $H_2$ concentration are controlled by surface emissions and deposition, production and loss by atmospheric chemistry and by atmospheric transport. Due to the slow response of relatively well mixed

H$_2$ in the atmosphere, with lifetime $\sim$2 yr (see Ehhalt and Rohrer, 2009; Patterson et al., 2020; Warwick et al., 2023; Sand et al., 2023), we assume that the general effect of these fluxes is well approximated when they are modelled in their zonal-mean monthly-mean.

We first introduce a prototype deposition scheme based on formulations of the moisture and temperature dependence of soil biology and diffusion processes (Sec. 2). Then we provide a new analysis of the observed H$_2$ distribution and seasonality (Sec. 3). We include the prototype depositions scheme into a toolbox model (Sec. 4) with estimates of: emissions with a spatial and monthly signal from Paulot et al. (2021), together with emissions strength and atmospheric chemical production and destruction fluxes from Sand et al. (2023); and tropospheric transport idealised as a latitude-height overturning from ERA5 monthly mean wind speeds (Hersbach et al., 2020) and atmospheric dispersion parameters tuned based on reproducing the observed SF$_6$ distribution. By comparing the simulated H$_2$ concentration against the observed distribution and seasonality we determine new constraints on the soil deposition as a function of latitude that we apply to tune the original prototype scheme (Sec. 5).

## 2 A Prototype Deposition Scheme

H$_2$ oxidising bacteria are ubiquitously distributed in soils (Schlegel, 1974; Khdhiri et al., 2015; Greening et al., 2016; Ji et al., 2017; Bay et al., 2021; Greening and Grinter, 2022).

Recently, Bertagni et al. (2021) formulated the uptake of atmospheric H$_2$ – constrained by the rate of gas diffusion into soil and its microbial activity – as functions of soil type, temperature and moisture, and without including a function of soil carbon, to derive a global model for H$_2$ deposition.

**Table 1.** Mapping of ERA5 soil textures to the Bertagni et al. (2021) soil parameters. The *Medium* and *Medium fine* textures dominate over the land surface.

| | ERA5 Soil Texture | Bertagni et al. (2021) Soil Texture |
|---|---|---|
| 1: | Coarse | Sand |
| 2: | Medium | Sandy loam |
| 3: | Medium fine | Silt |
| 4: | Fine | Sandy clay |
| 5: | Very fine | Clay |
| 6: | Organic | Loam |
| 7: | Tropical Organic | Loam |

To simplify our analysis of the dominant drivers of seasonality in H$_2$ deposition we implement a prototype scheme that isolates the deposition rate seasonality due to terms limited by soil moisture, $s$, proportional to $f(s)$ and $g(s)$ and by soil temperature, $T$, proportional to $h(T)$ (Ehhalt and Rohrer, 2011; Ehhalt and Rohrer, 2013; Bertagni et al., 2021) . The terms $f(s)$ and $h(T)$ limit potential microbial activity and $g(s)$ models the gas diffusivity into the soil. Ehhalt and Rohrer (2013) and Bertagni et al. (2021) identified the importance of high-frequency fluctuations in their deposition models – particularly as

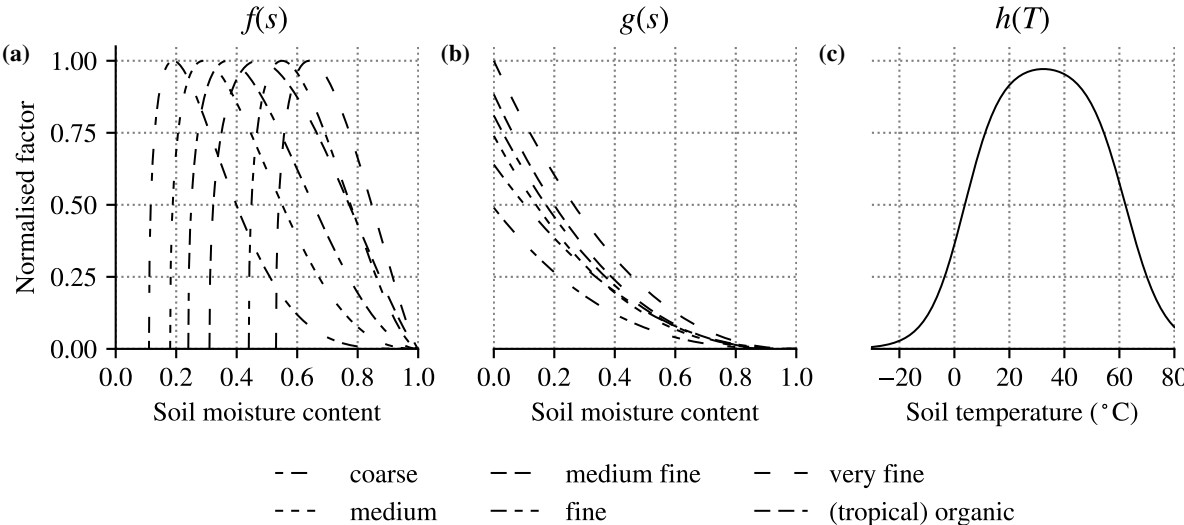

**Figure 1.** Normalised soil deposition rate factors (sourced from Ehhalt and Rohrer, 2011, Bertagni et al., 2021): (**a**) $f$ and (**b**) $g$ for different soil textures used in ERA5 (Table. 1); and (**c**) $h$.

a product of the changing soil moisture depth-profile through cycles of precipitation and drying. As we base our constraints on the planetary scale seasonality at lower frequencies of 1-2 $\mathrm{yr}^{-1}$, we drive $f$, $g$ and $h$ with monthly mean ERA5 soil moisture in the top 7 cm and skin temperature, and scale the deposition rate with a constant to close the $H_2$ budget as in Sand et al. (2023) (summarised in Table. 3). A comparison of $f$ and $g$ with soil moisture varying on 30-day and daily timescales is provided in

Fig. A1. The use of monthly average ERA5 data can lead to slightly faster uptake rates in semi-arid regions (cf. Bertagni et al., 2021) but does not substantially affect the seasonality.

Analytically, the resultant $H_2$ uptake is the parallel sum of the potential flux limited by biological activity with the potential flux limited by diffusion (Bertagni et al., 2021). However, that formulation would require accurate quantification of each flux. Alternately, Ehhalt and Rohrer (2013) have formulated deposition velocity in moist soils to vary with $(fgh)^{1/2}$. However, in

this work, the objective of the prototype scheme is to capture the seasonality in these processes while facilitating an analysis of how these processes drive the planetary $H_2$ distribution. Therefore, we adopt an idealised formulation where the total uptake is proportional to the product of the normalised terms, $fgh$, and is scaled to achieve a total 57.2 Tg $\mathrm{yr}^{-1}$ average deposition (following Sand et al., 2023).

Suppressed $H_2$ uptake has been measured in soils at low and high moisture contents (Conrad and Seiler, 1981), yet Bertagni

et al. (2021) emphasise the continued lack of quantitative observations for how soil biological activity varies with soil moisture. In lieu of this data, they provide an adaptable model which constrains the soil moisture limiting function with the soil matric potential for different soil textures. Accepting this persistent difficulty, we define $f(s)$ (Fig. 1a) as a mapping of that defined function to the ERA5 set of soil textures (Table. 1).

Bertagni et al. (2021) identify the importance of a diffusive limitation on $H_2$ deposition in humid regions. They model the $H_2$ flux into the soil proportional to the parallel addition of the gas-conductance of the diffusive soil layer, $g$, with the gas-conductance of any diffusive barrier – such as organic litter or snow cover. They adopt the soil structure dependent gas diffusivity model of Moldrup et al. (2013) from which we propose an idealised moisture dependent gas conductivity for $H_2$ into soil, assuming a constant free-air diffusion rate and diffusive layer length-scale,

$$g(s) \propto n^2 (1-s)^{2+\frac{3}{b}} \tag{1}$$

where the soil porosity $n$ and the parameter $b$ are compiled for different soil textures in Bertagni et al. (2021). The resulting normalised $g$ is plotted in Fig 1b.

The effects of diffusive barriers are diagnosed by Bertagni et al. (2021). Their presence has the strongest limiting effect where the underlying soil diffusivity is highest but this is masked where the biological uptake is strongly limited due to lack of soil moisture, or by cold temperatures where there is snow cover. Therefore, rather than relying on broad assumptions for the distribution of diffusive barriers, in our prototype scheme we assume a total gas diffusivity proportional to $g$.

We choose $h(T)$ as the normalised soil temperature dependence across $H_2$ removal experiments that was defined by Ehhalt and Rohrer (2011) (Fig. 1c), where $H_2$ removal occurs from temperatures as low as $-20°$C, increases following a Fermi distribution to a peak at around $30°$C (cf. Smith-Downey et al., 2006), and is limited quickly for temperatures higher than $40°$C.

## 3 Filtering and Decomposition of H$_2$ Observations

Timeseries of H$_2$ mixing ratios (Pétron et al., 2023) have been measured at the NOAA Global Monitoring Laboratory from flask samples received from a latitude spanning network of sites (NOAA Global Monitoring Laboratory, 2024). These flask samples have typically been taken once or twice weekly since 2010 (Pétron et al., 2023) using a portable sampling unit with a c.5m mast, and samplers are instructed to preferably sample upwind of buildings at times when wind speeds are $\geq 2$ ms$^{-1}$ (NOAA Global Monitoring Laboratory, 2005).

We apply a spatial filtering to the measurement sites based on the predicted annual mean deposition rate in the biophysics based prototype deposition scheme that we introduce in Sec. 2 (Fig. 2a). By drawing a systematic comparison across sites we are able to extract general features of the planetary signal of background surface H$_2$ distribution and seasonality.

Additional to the spatial filtering of sites, temporal filtering is used to decompose how the observed H$_2$ signal varies over different timescales. We analyse the seasonality at each site then compare these signals across latitudes rather than finding the seasonality of latitude-clustered sites (cf. Paulot et al., 2024), restricting our analysis to a set of station measurements where consistent sampling exists in the period 2012-2018.

We implement a filter $F_{mid}$ to isolate the seasonal cycle. $F_{mid}$ returns the recorded data minus the high-frequency noise isolated with a high-pass filter, $F_{high}$, and the inter-annual variation isolated with a low-pass filter, $F_{low}$. This decomposition is illustrated for data recorded at the Mace Head atmospheric research station in Fig. 3. High-frequency noise driven by synoptic

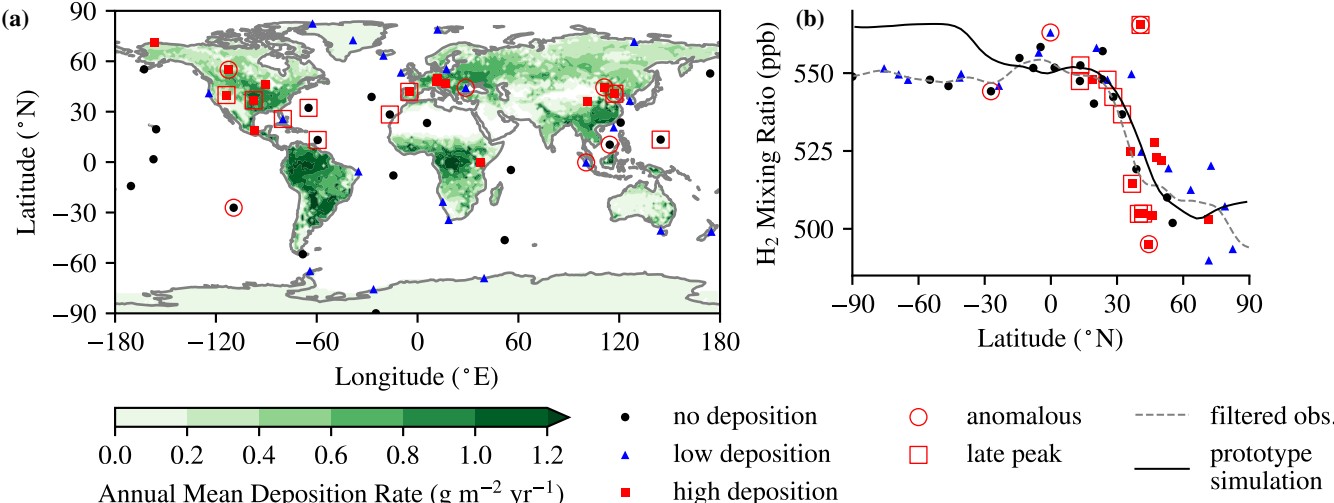

**Figure 2.** (**a**) Annual mean $H_2$ deposition flux (g m$^{-2}$ yr$^{-1}$) in the prototype deposition scheme (shading), with surface $H_2$ measuring stations (symbols) (NOAA Global Monitoring Laboratory, 2024) filtered by local deposition in the prototype deposition scheme (Table 2). (**b**) observed 2012-2018 mean $H_2$ mixing ratio at surface measuring sites (symbols, labelled according to the average prototype deposition within $\pm 2.5°$N and $\pm 2.5°$E: no deposition; low deposition $\in (0, 0.2]$ g m$^{-2}$ yr$^{-1}$; and high deposition $> 0.2$ g m$^{-2}$ yr$^{-1}$.) (Pétron et al., 2023) and the near-surface zonal model results from a simulation using the prototype deposition scheme (solid line). Anomalous sites (circled) are identified following Equation B1. A cluster of sites in the NH subtropics have a peak in the first harmonic after 150 days (squared, in Fig. 4 peak 1-2 months later than other sites in the sub-tropics). Filtered observations (grey dashed line) is a fit to the observational data excluding anomalous stations (circled) using a Gaussian filter with $\sigma = 5°$lat. Plotted station data are summarised in Table. C1.

weather occurs on timescales $< 30$ days and is isolated with $F_{high}$ by subtracting a central moving average with a 30 day window from the data. Subtracting the high-frequency noise from the data reveals a background state where the remaining temporal variation is dominated by a seasonal cycle (Paulot et al., 2024). Therefore, to isolate the low-frequency inter-annual variability and trends it is suitable to define $F_{low}$ as a central moving average with a 1 year window.

This background state reveals the trend of increasing atmospheric burden varying on timescales $\tau_{low} \geq 1$ year (Novelli, 1999; Patterson et al., 2020; Ehhalt and Rohrer, 2013). Furthermore, the meridional gradient of its mean state (Fig. 2b), combined with quantification of the atmospheric chemical production and loss of $H_2$, reveals the net surface fluxes in either hemisphere (Sanderson et al., 2003).

    The first harmonic of the seasonality at each site (Fig. 4a,b) is identified by optimising the fit of the curve

$$h_1(A_1, \Phi_1) = A_1 \cos\big(\Omega(t - \Phi_1)\big) \tag{2}$$

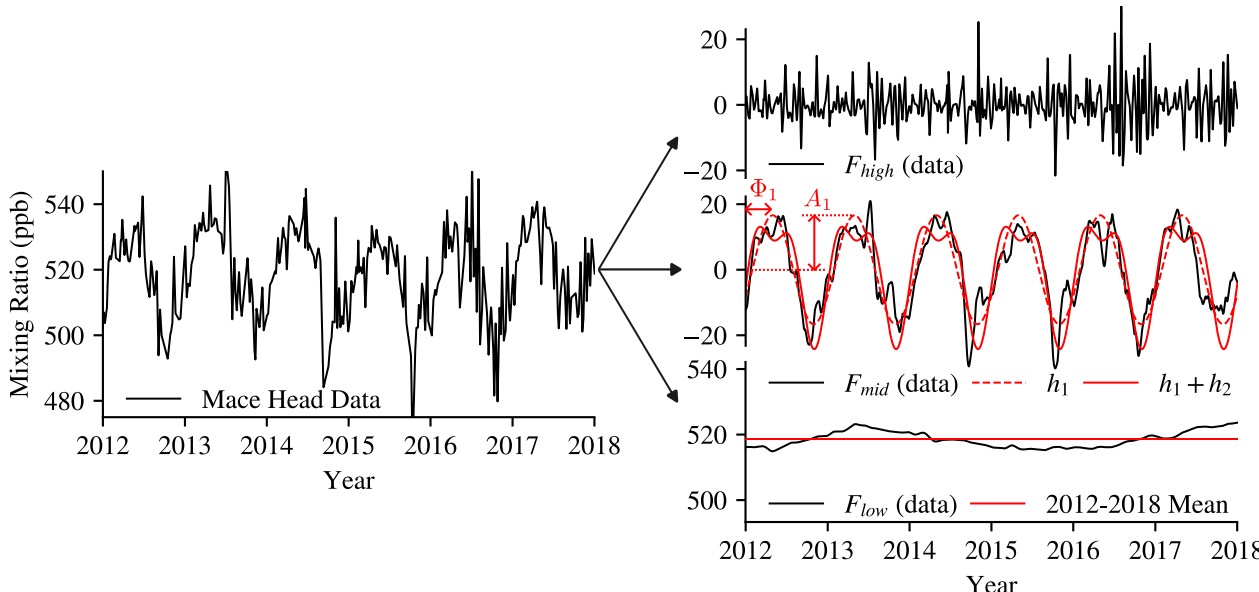

**Figure 3.** Example decomposition of the 2012-2018 $H_2$ observations from the Mace Head atmospheric research station (left) (Pétron et al., 2023) into: top-right, high-frequency noise on timescales $< 30$ days ($F_{high}$); mid-right, seasonality on timescales 30 days - 1 year ($F_{mid}$), with first ($h_1$) and second ($h_2$) harmonics fitted to this (red solid and dashed lines respectively); bottom-right, the residual inter-annual variation ($F_{low}$); and the 2012-2018 mean (red line). The amplitude, $A_1$, and phase, $\Phi_1$, of the first harmonic are indicated.

to the $F_{mid}$ filtered data, where $A_1$ and $\Phi_1$ are, respectively, the amplitude and phase of the harmonic (illustrated for Mace Head in Fig. 3), and $\Omega = 2\pi \text{ yr}^{-1}$. Additionally, a second harmonic of the seasonality is identified by optimising the fit of

$$h_2(A_2, \Phi_2) = A_2 \cos\big(2\Omega(t - \Phi_2)\big) \tag{3}$$

to the $F_{mid}$ filtered data minus $h_1$ (Fig. 4c,d).

The accuracy of describing the seasonality with $A_1, \Phi_1, A_2, \Phi_2$ at each site is assessed by considering the root-mean-square (RMS) error of $h_1 + h_2$ minus the $F_{mid}$ filtered data. Anomalous results are categorised where this RMS error is greater than 20 ppb (Equation B1, sites indicated in Fig.2). By inspection, this criterion effectively excludes five sites where harmonics could not be identified (such as in Fig. 3).

    Between 2012 and 2018 both the annual mean surface $H_2$ mixing ratio (Fig. 2b) as well as the amplitude (Fig. 4a) and
phase (Fig. 4b) of the seasonal variability – the time of the peak of the first harmonic of the seasonal variability – are well described with a function of latitude. A Gaussian filter with a standard deviation of $5°$ latitude is used to find a best-fit between observations excluding anomalous sites. The small spread of the phase of the seasonality of $H_2$ observations from the best-fit shows that the mid-filtered signal varies more with the latitude of sites than due to zonal variations in local deposition. We exploit a $H_2$ seasonality that continuously varies with latitude as a distinct constraint compared to Paulot et al. (2024). This
constraint allows us to test assumptions in the prototype deposition scheme at higher meridional resolution (Sec. 5).

Neglecting the trend of increasing $H_2$ concentrations – which are propagated to the planetary scale due its long atmospheric lifetime – and considering the dominant role of surface uptake in the $H_2$ sink, this meridional gradient (Fig. 2b) suggests a net down gradient mixing from the tropics to the NH high-latitudes contrasted with approximately no meridional gradient between the tropics and SH mid- and high-latitudes. This supports the assumption that the deposition into soils is dominated by the larger land area of the NH, where this soil sink exceeds anthropogenic emissions and the net source of $H_2$ from atmospheric chemistry (Paulot et al., 2021).

Outside the tropics, the first harmonic of the seasonal oscillation peaks in the late spring to early summer – between April and June – in the NH, and in late summer – late February – in the SH (Fig. 4b). The amplitude of this harmonic increases with latitude in the NH, and is reduced in the deep tropics compared with the subtropics, reflecting the different seasonality in the tropics compared with temperate regions (Fig. 4a). Figure 4b shows a spread in the phase of the seasonality in the NH between 10 and 45°N of about 2 months, from late April to late June. There are a cluster of measurements where the first harmonic peaks in June – later in summer in the NH, more similar to the late summer peak in the SH (red squares, Figs. 2,4). These sites are spread zonally, and have an amplitude of this first harmonic, as well as a second harmonic, of seasonality consistent with other observations in the NH (Figs. 4a,c,d).

We define a reference monthly mean $H_2$ mixing ratio, $r_{ref}$, as the sum of the observed annual mean state and first and second harmonics fit to the observations with a Gaussian filter with $\sigma = 5°$lat excluding stations with anomalous signals. The width of the filter was chosen to produce a smooth best-fit between observations that preserves the distinct patterns with latitude that vary over scales c.10°N. This reference $H_2$ signal isolates the dominant signal from noise and inter-annual variability, and the decomposition of $r_{ref}$ recovers the best-fit to the 2012-2018 surface observations: in the annual mean (grey dashed, Fig. 2b); the first harmonic of the seasonality (grey dashed, Figs. 4a,b); and the second harmonic (grey dashed, Figs. 4c,d).

The importance of deposition seasonality is indicated in Fig. 4: a simulation with the same annual deposition flux, but without deposition seasonality, (dotted line) fails to reproduce key features of the planetary $H_2$ seasonality. In the simulation with $H_2$ seasonality driven by emissions, deposition and atmospheric chemistry (solid line), in both hemispheres $H_2$ peaks in late-summer to early-autumn – February-March in SH and August-September in NH – with similar zonal-mean peak amplitude 10-15 ppb to the observations in the mid-latitudes. Seasonally varying deposition is required for NH $H_2$ to peak earlier in the year and to resolve the distinct latitude bands of peak seasonality. Alternately, prototype deposition seasonality has little impact in the SH, due to the relative lack of land. In the SH, the seasonality of $H_2$ is mainly controlled by atmospheric chemistry.

## 4   Model Formulation

Analysis of the 2012-2018 observational data for $H_2$ mixing-ratio (from Pétron et al., 2023) showed that the mean and first and second annual harmonics of the $H_2$ distribution (Figs. 2b and 4a,b,c,d) are well described with a function of latitude. Additionally, we note the long average lifetime of $H_2$ compared with timescales of horizontal mixing in the atmosphere (e.g. Pierrehumbert and Yang, 1993) indicating that $H_2$ is reasonably well mixed across zonal bands. However, this assumption is

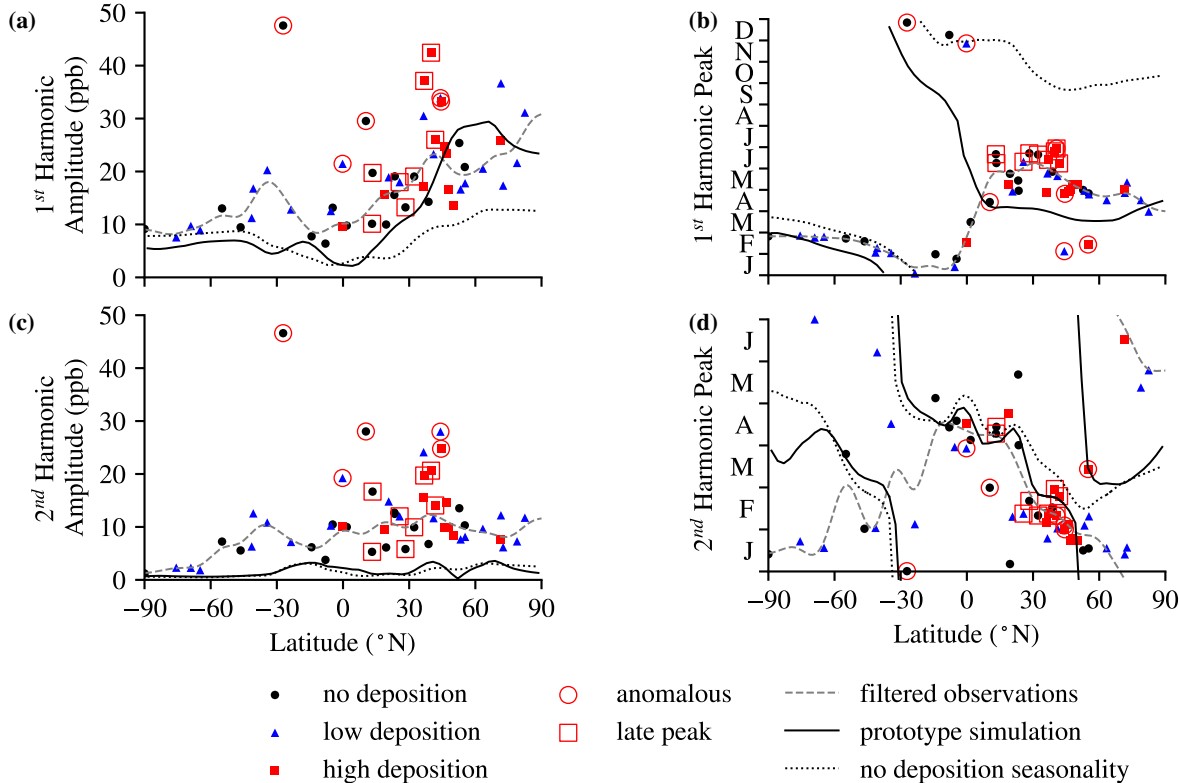

**Figure 4.** Seasonality of the 2012-2018 $H_2$ mixing ratio at surface measuring sites (symbols, see Fig. 2) (Pétron et al., 2023), the near-surface conditions of a simulation using the prototype deposition scheme (solid line, see Table. 2) and using the annual mean deposition velocity of this scheme (dotted line): (**a**) and (**c**) the amplitudes of the first and second harmonics of the seasonal oscillation ($A_1$ and $A_2$); and (**b**) and (**d**) the phase of the peak of the first and second harmonics of the seasonal oscillation ($\Phi_1$ and $\Phi_2$). Note, the pattern for $\Phi_2$ over the first six months repeats over the last six months. Filtered observations (grey dashed line) is a fit to the observational data excluding anomalous stations using a Gaussian filter with $\sigma = 5°$lat.

**Table 2.** Summary of deposition schemes used and devised in this study.

| Deposition Scheme | Description | Experiments |
|---|---|---|
| prototype scheme | proportional to the leading terms in Ehhalt and Rohrer (2013) and Bertagni et al. (2021): $W_{prototpye} = k\,f(s)\,g(s)\,h(T)$, where $k$ is a constant to scale annual deposition to the multi-model mean from Sand et al. (2023) | prototype simulation (Figs. 2,4,7,8,10) |
| best-fit 1 (BF1) | deposition anomaly that reproduces observed seasonality: found as an anomaly from a simulation with the annual mean prototype deposition scheme as a basic state | BF1 (Figs. 7,8) |
| best-fit 2 (BF2) | prototype scheme perturbed with an anomaly: an anomaly is found from a simulation with the monthly varying prototype scheme as the basic state | BF2 (Figs. 9,10) |

challenged at latitudes with particularly strong sources and sinks. For example, there is a spread of c.5% in the non-anomalous observed annual mean mixing ratios around 40°N (Fig. 2b).

Therefore, we attempt to simulate a planetary $H_2$ signal in a simple 2D (latitude-height) model with monthly-varying zonal-mean emissions, atmospheric loss and production, soil deposition, and transport integrated with a $4^{th}$ order Runge-Kutta method, with 30-day months and 4 steps per day. The latitude-height model comprises 64 equally spaced latitude bands with 3-layers representing the lower, middle and upper troposphere, for which we assume fixed pressure boundaries at 1000, 800, 600 and 150 hPa. The simplicity of the model is prioritised such that simulations carry a low computational cost and differences

resulting from the model configurations may be readily identified.

## 4.1 Emissions of $H_2$

We constrain the total $H_2$ emissions and production and destruction by atmospheric chemistry to match the multi-model average fluxes estimates in Sand et al. (2023) from six models driven by prescribed boundary layer $H_2$ and $CH_4$ concentrations (summarised in Table. 3). In Sand et al. (2023) total $H_2$ emissions were estimated as the residual in offline calculations consid-

ering simulated atmospheric $H_2$ production and destruction, and estimated $H_2$ deposition. They found in an inter-model mean of $35.7 \pm 16.3$ Tg yr$^{-1}$ (summarised in Table. 3 from Sand et al., 2023), which corresponds with the estimate of 1995-2015 average $H_2$ emissions of Paulot et al. (2021): 29.9-37.1 Tg yr$^{-1}$.

We set annual $H_2$ emissions to $35.7 \pm 16.3$ Tg, of which 7.8 Tg are monthly varying biomass burning emissions with the *input4MIPs* estimate (Marle et al., 2017). The remaining emissions from anthropogenic sources and nitrogen fixing are

205 implemented using the monthly mean signals from Paulot et al. (2021) in their respective proportions, but requiring a c.$+20\%$ scaling: 17.1 Tg and 10.8 Tg respectively. The seasonality and latitudinal distribution of emissions are shown in Fig. 5a,b.

## 4.2 Atmospheric Chemistry of $H_2$

As shown in Figs. 2b and 4 the prototype simulation (black line) captures much of the observed $H_2$ distribution and seasonal variation (symbols) at the surface, and due to the small amplitude of seasonal variability compared with the annual mean $H_2$

mixing ratio, the $H_2$ concentrations in the initial prototype simulation and simulations that achieve a best-fit to observations will agree to within a few percent. While in reality loss fluxes are proportional to the $H_2$ mixing ratio, the close agreement permits a choice of atmospheric production and loss fluxes that do not depend on the $H_2$ mixing ratio in each simulation: the fluxes for production and destruction by atmospheric chemistry are taken from a simulation with the UKCA model with imposed $H_2$ mixing ratios at the boundary layer used in Sand et al. (2023). These fluxes are close to the multi-model mean

in that study, but are scaled by a small amount such that the fluxes used in this study align with the atmospheric $H_2$ budget summarised in Table. 3. The vertical sum of net chemical fluxes are shown in Fig. 5c.

## 4.3 Atmospheric Transport of $H_2$

An idealised transport scheme is implemented as a monthly mean overturning in the troposphere and an empirically tuned representation of mixing. To ensure mass conservation in the overturning scheme, we first calculate a streamfunction from

220 the monthly mean meridional overturning in ERA5 data over the period 2010-2020 (see Hersbach et al., 2020). Idealised representations of horizontal and vertical mixing as constant rates between layers and zonal bands are tuned to reproduce the meridional gradient in $SF_6$ from observations in the *World Data Centre for Greenhouse Gases* dataset from di Sarra et al. (2023) for simulations driven with *Emissions Database for Global Atmospheric Research* $SF_6$ emissions from Crippa et al. (2023).

**Table 3.** Summary of annual mean $H_2$ fluxes input to the model from: † Sand et al. (2023) mean over six fixed boundary layer concentration driven models with estimated emissions and a standard-deviation over the six models; ∗ the emissions from biomass burning of the total emissions from Marle et al. (2017).

|  | Mean (Tg yr$^{-1}$) | Inter-Model $\sigma$ (Tg yr$^{-1}$) |
|---|---|---|
| Atmosphere Production † | 46.8 | 7.4 |
| Atmosphere Loss † | $-25.2$ | 3.2 |
| Estimated Total Emissions † | 35.7 | 16.3 |
| Biomass Burning Emissions ∗ | 7.8 | - |

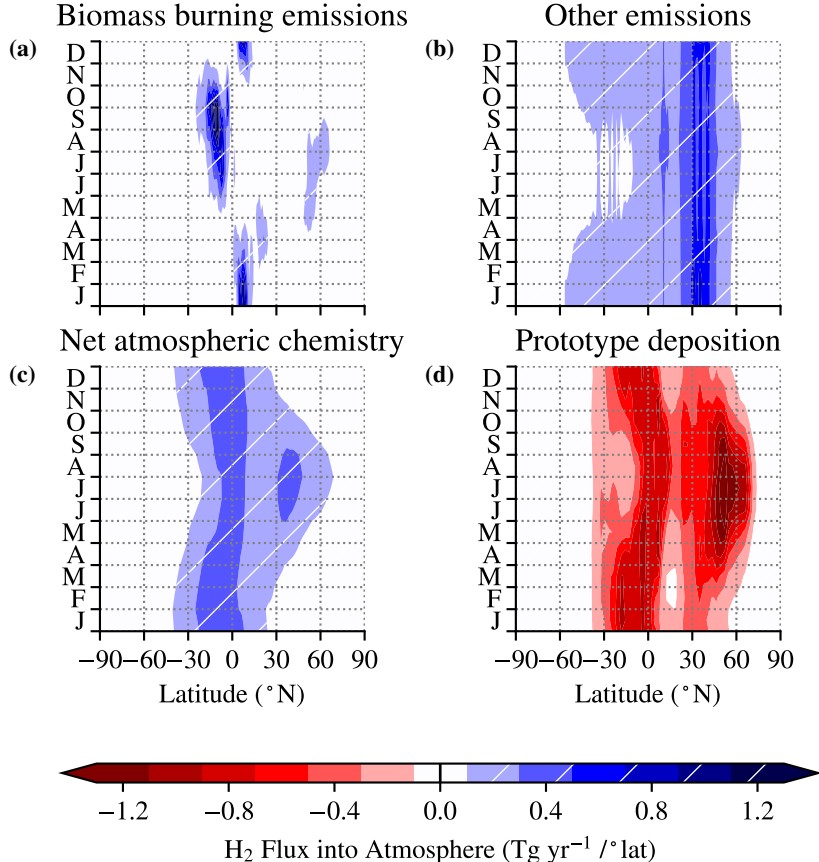

**Figure 5.** $H_2$ fluxes in the model: (**a**) emissions from biomass burning (from Marle et al., 2017); (**b**) other emissions including from combustion from fossil fuels and nitrogen fixing in soils and oceans (based on Paulot et al., 2021; Sand et al., 2023); (**c**) net production minus destruction by atmospheric chemistry (from Sand et al., 2023); and (**d**) zonal mean deposition with prototype scheme.

## 5 A Best-Fit Deposition Scheme

The remaining $H_2$ flux in the model is through deposition to soils – plotted for a simulation with the prototype deposition scheme in Fig. 5d. The prototype deposition scheme and each term may be decomposed into an annual mean state and a seasonality,

$$fgh = \overline{fgh} + (fgh)', \tag{4}$$

$$fg = \overline{fg} + (fg)', \tag{5}$$

$$h = \overline{h} + h', \tag{6}$$

where the over-bar refers to an annual mean, spatially varying field, and dashed fields are the seasonality. Figure 6 shows that the coefficient of variation – the ratio of the temporal standard deviation to the mean at each latitude – of $f$, $g$ and $h$ are distinct functions of latitude, such that throughout most latitudes either $|(fg)'| \ll \overline{fg}$ or $|h'| \ll \bar{h}$ wherever there is strong seasonality in the other term. This result shows that in most cases the cross-term $|(fg)'h'| \ll \overline{fgh}$, such that

$$\overline{fgh} \approx \overline{fg}\,\bar{h}, \tag{7}$$

and as a consequence,

$$(fgh)' \approx \left(\overline{fg}h(T) - \overline{fgh}\right) + \left(f(s)g(s)\bar{h} - \overline{fgh}\right), \tag{8}$$

where the seasonality due to variations in soil moisture and the seasonality due to variation in soil temperature may be separated. The seasonality of the deposition flux is then calculated as the product of the seasonality of the zonally integrated deposition velocity and $r_{ref}$, the signal of observed H$_2$ mixing ratios. This is shown in Fig. 7a.

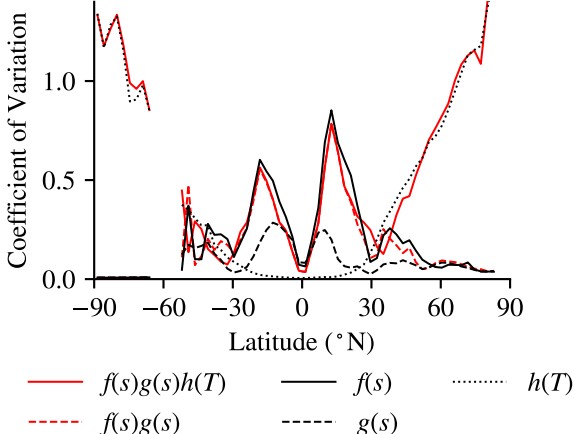

**Figure 6.** Coefficient of seasonal variation for the zonal mean prototype deposition scheme (solid red); the factors that vary with soil moisture (dashed red); and terms $f$ (solid), $g$ (dashed) and $h$ (dotted).

An anomaly to the net surface flux is constructed of a perturbation to the prototype emissions and a perturbation to the prototype deposition scheme. We first identify the latitude-time signal of a deposition scheme that captures the best-fit to the seasonality of the observations independent of the seasonality of the prototype deposition scheme. This is the best-fit anomaly, BF1 (Table. 2), required to reproduce $r_{ref}$ from a model with a basic deposition state taken as the annual mean deposition velocity of the prototype scheme, $\overline{fgh}$. BF1 is calculated as the average rate of relaxation of a number of small Newtonian relaxations, $R$, towards $r_{ref}$ through each month of integration. This is found in an inverted version of the model, where the H$_2$ mixing ratio in the lower layer, $r$, changes as

$$\frac{\partial}{\partial t}r = S + M - r\overline{fgh} - R, \tag{9}$$

where

$$S = E + P - rD \tag{10}$$

represents emissions, $E$, and atmospheric chemistry production, $P$, and destruction, $rD$, $M$ represents mixing, and the relaxation term

$$R = \delta_R(r - r_{ref}), \tag{11}$$

where $\delta_R \ll 1$ is a constant chosen such that $r$ relaxes sufficiently close to $r_{ref}$ by the end of the month, but allows steady adjustment and mixing of $r$ over that period. Note, in this experiment, in the lower layer the seasonality in $r$ is much smaller than the annual mean such that $|r - r_{ref}| \ll \bar{r}$, and $\sim 70\%$ of the total $H_2$ sink is by deposition into the soil (Sand et al., 2023) such that $rD \ll r\overline{fgh}$, the net chemistry is assumed to be captured with the same monthly varying flux in each test.

We compare BF1 (Fig. 7b) to the seasonality of the prototype deposition scheme and its decomposed terms (Figs. 7a,c,d). The seasonality of the prototype scheme, $(fgh)'$, reproduces some key features of BF1. In particular, the prototype scheme captures the strong seasonality of BF1 in the NH mid-latitudes. There, seasonality is driven by the sensitivity of microbial activity to variation in soil temperature, which has been identified in laboratory microbial studies (Smith-Downey et al., 2006). Alternately, the seasonality driven by soil moisture changes is dominant in the tropics.

To analyse how adjustments to the seasonality of the prototype deposition scheme affect results we define $R_{BF}(\alpha, \Delta t)$ as the ratio of the RMS error between the seasonality of a deposition scheme – adjusted by scaling the seasonality of the prototype scheme by a factor $\alpha$ and offsetting in time by $\Delta t$ – and BF1 to the RMS seasonality of BF1 at each latitude (Equation D1). This ratio indicates how well the adjusted deposition scheme performs at reproducing the best-fit deposition scheme: $R_{BF} = 0$ occurs where the adjusted scheme locally reproduces the best-fit scheme; $R_{BF} = 1$ occurs if the adjusted scheme performs as well as the annual mean prototype deposition scheme, $\overline{fgh}$; and $R_{BF} > 1$ indicates that the adjusted scheme performs worse than $\overline{fgh}$.

In Fig. 8a the seasonality of $fgh$ only substantially reproduces BF1 where the strength of its seasonality is decreased in the temperate NH between 45-70°N. Alternately, in Fig. 8b, $fgh$ better reproduces BF1 scheme when it includes a lag of two to four months in the tropics, or half a month later in the NH mid-latitudes.

In Fig. 8c the latitudes of peak amplitude of seasonality in the prototype scheme are isolated for optimised agreement with BF1 under adjustments to the seasonality multiplier and offset. In both deep tropical peaks (A,B), better agreement occurs for a lag of two to three months. Additionally, the strongest agreement occurs for a weaker seasonality in the SH tropical peak (A). In the NH mid-latitudes better agreement is achieved with a one week lag in the peak deposition at 52°N (C).

Figure 8c shows that $R_{BF}$ is minimised for decreases in the amplitude of the seasonality at A, B, C. However, this arises as BF1 is unconstrained by the seasonality in the prototype scheme, instead assuming variations in deposition spread across latitudes. By integrating the deposition seasonality across wider latitude bands: Figs. 8d,e show a lag and increases in amplitude of seasonality between 30°S and 30°N; whereas Fig. 8f shows differences in the seasonal signal in the NH mid-latitudes, but with a similar amplitude (cf. Figs.7a,b). The double peak in the deposition seasonality of BF1 in the tropics (Figs. 8d,e) suggests

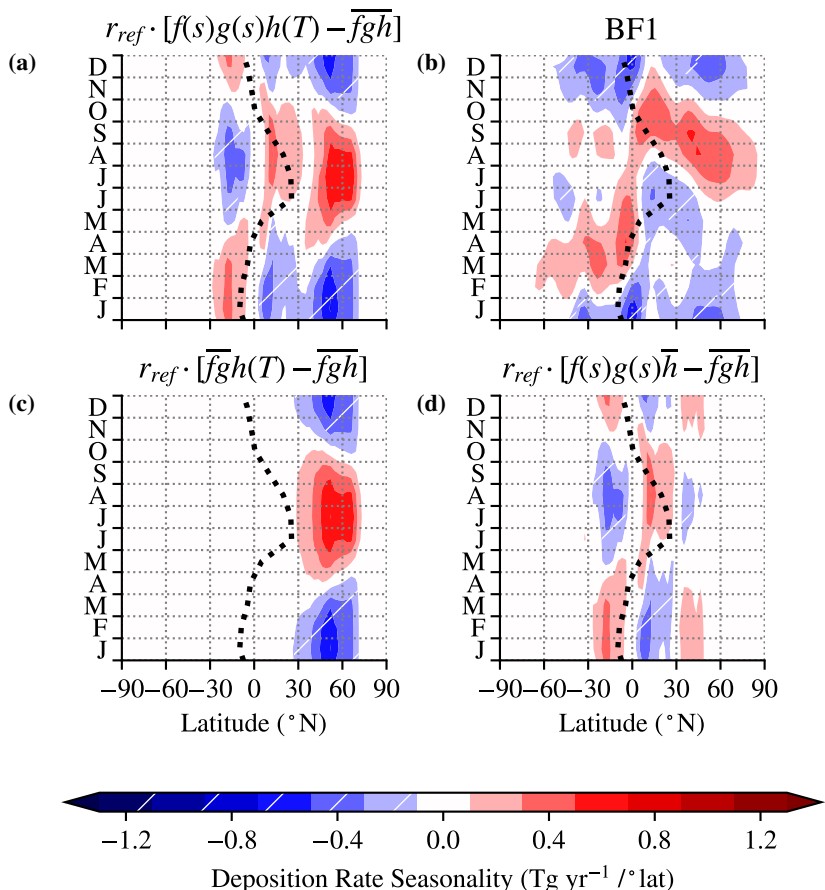

**Figure 7.** Deposition rate seasonality (shaded) for: (**a**) the prototype deposition scheme multiplied by near-surface mixing ratios of this simulation; (**b**) BF1 from $\overline{fgh}$ found in the inverted model; Panels **c** and **d**) the prototype deposition scheme with isolated monthly variation that depend only on $h$ and $fg$ respectively (Equation 8). In each panel the ITCZ migration (black dots) is included as the order 10 precipitation centroid (from ERA5 2010-2020) between 20°S and 20°N following Adam et al. (2016).

that the fastest uptake may occur coincidentally with the ITCZ crosses the equator at the equinoxes (Fig. 8) rather than when the ITCZ is furthest from the equator, as in the prototype scheme. This implies that BF1 may reflect a soil moisture interaction not captured in the prototype scheme. Paulot et al. (2024) have recently shown how a deposition scheme driven by three-hourly varying soil parameters from the Global Land Data Assimilation System (Rodell et al., 2004), and a low soil moisture activation threshold for bacterial uptake, produced a double-peak in $H_2$ in the tropics and captured NH $H_2$ seasonality. This was distinct from their base simulation driven by monthly-mean soil moisture, where NH sub-tropical $H_2$ peaked three months earlier than observations, comparable with the prototype scheme driven with monthly-mean ERA5 data in this work (Fig. 4b).

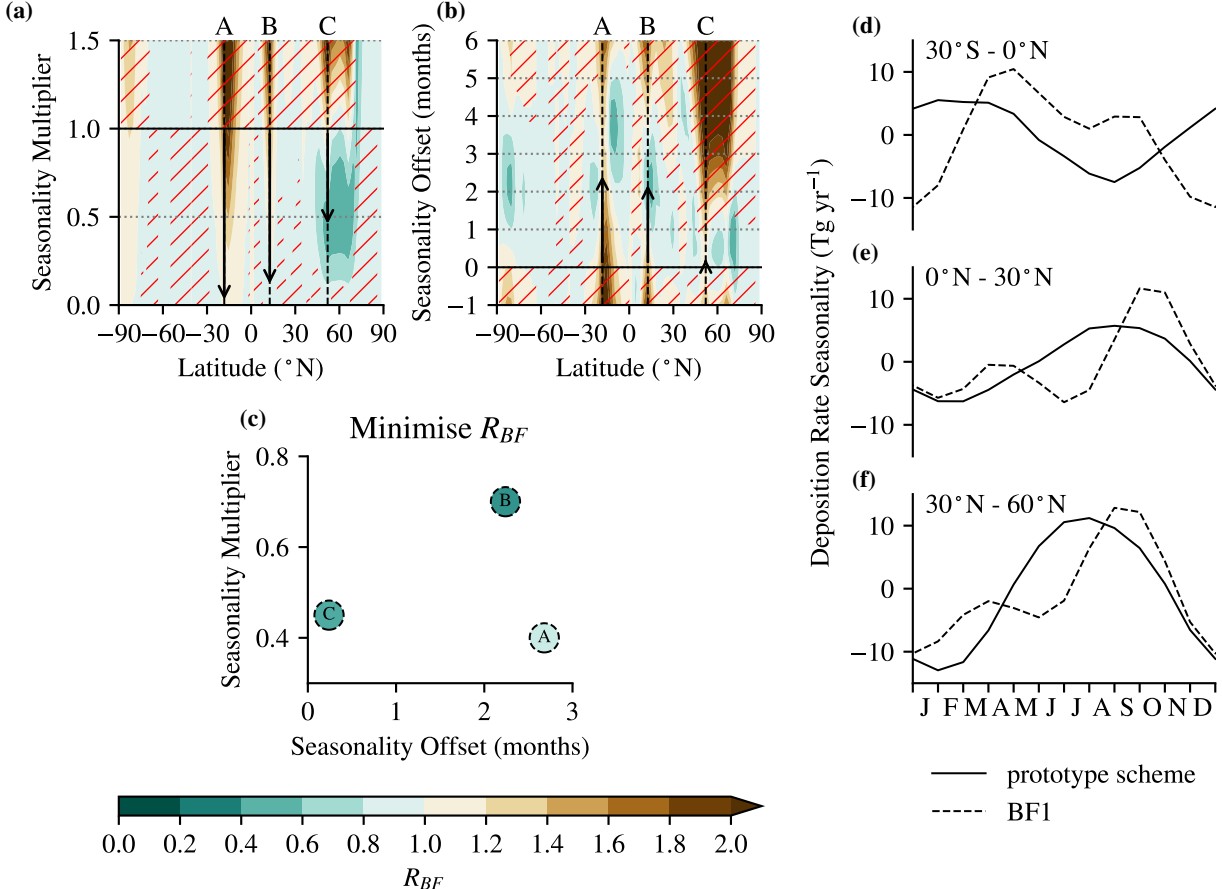

**Figure 8.** (**a**-**c**) Colours show $R_{BF}$ (Equation D1), which measures the performance of adjusted versions of the prototype deposition scheme at reproducing the best-fit deposition scheme, BF1. $R_{BF} = 1$ when the adjusted scheme performs as well as annual mean deposition with no seasonality, and $R_{BF} = 0$ when the deposition scheme reproduces BF1. In (**a**), only the amplitude of the deposition seasonality is adjusted, by scaling with a multiplier, $\alpha$. In (**b**), only the timing of the deposition seasonality is adjusted, by including an offset, $\Delta t$. Arrows indicate optimal adjustments at the latitudes of peaks in deposition seasonality in the prototype scheme (annotated A:18°S, B:13°N and C:52°N in each panel, see Fig. 7a) when $\alpha$ and $\Delta t$ are adjusted individually. (**c**) the optimal RBF achieved at latitudes A, B and C when $\alpha$ and $\Delta t$ are adjusted jointly. Seasonality of deposition for the prototype and BF1 schemes, integrated across three latitude bands: (**d**) 0-30°S; (**e**) 0-30°N; and (**f**) 30-60°N.

## 6 Contribution from Emissions and Chemical Production and Destruction

We have shown that the seasonality of the monthly varying prototype deposition scheme, $fgh$, captures the key features of a scheme that represents a best-fit for the observed seasonality independent of the seasonality of the prototype scheme, BF1. Given that similar seasonal signals to the biophysics based seasonality of the prototype scheme are independently reproduced by BF1 we examine a second best-fit scheme, BF2 (Table. 2), derived as a perturbation from the full monthly varying prototype scheme. In this case, Equation 9 becomes

$$\frac{\partial}{\partial t}r = S + M - rfgh - R, \tag{12}$$

where $S$ in unchanged but the mixing, $M$, and relaxation, $R$, terms respond to the change $\overline{fgh} \to fgh$. Like BF1, BF2 is calculated from the flux anomaly (Fig. 9a) contained in $R$ and required to resolve $r_{ref}$.

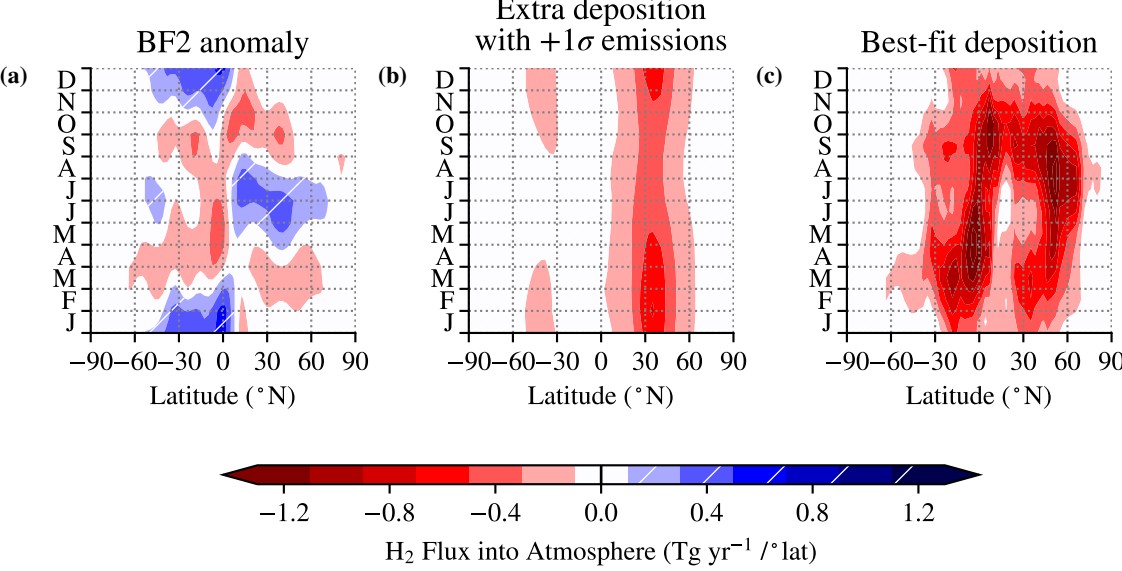

**Figure 9.** Different $H_2$ fluxes into the model: **(a)** BF2 anomaly into lower layer to reproduce observations versus a simulation using the prototype deposition scheme; **(b)** anomaly change per $+1\sigma$ change in 'other emissions'; and **(c)** 'best-fit deposition' under the prototype deposition scheme perturbed by BF2.

Due to the large inter-model spread in estimated $H_2$ emissions (Table. 3, Sand et al., 2023), it is necessary to consider how much of the BF2 anomaly may be explained by the assumed emissions (Figs. 5a,b) and chemistry (Fig. 5c) schemes. The

BF2 anomaly is strongest in the tropics and subtropics with a net upwards flux peaking south of the Equator in November-December and spreading into the northern tropics during the NH summer. This upwards flux is similar to the source from biomass burning (Fig. 5a) displaced several months later, where other studies have used stronger emissions for biomass burning than the inputs4MIPS (Marle et al., 2017) scheme (e.g. Novelli, 1999; Sanderson et al., 2003; Price et al., 2007; Xiao et al.,

2007). However, it also captures a weakening of the high deposition in the prototype deposition scheme driven by increases in $f$ following the migration of the ITCZ (Fig. 7d). Combined with the stronger deposition during the winter in the subtropics in both hemispheres, this structure captures an offset in the prototype deposition scheme of around 3 months as seen earlier for BF1 (Fig. 8c).

Alternatively, the upwards flux component of this anomaly may partly be explained through an intensification of emissions due to nitrogen fixing or net production by chemical processes during the summer in the subtropics and tropics. Increasing the intensity of the non-biomass burning $H_2$ emissions by one standard deviation of the Sand et al. (2023) inter-model emissions estimate requires enhanced deposition with a maximum of around 0.5 Tg yr$^{-1}$/°lat focused in the NH mid-latitudes, and peaking in both hemispheres between December and February (Fig. 9b).

## 7 Effect on $H_2$ Lifetime of the Perturbed Scheme

To examine the effect of changing the prototype deposition scheme to the best-fit deposition scheme, BF2, we conduct a series of $H_2$ perturbation experiments. To gauge a sensitivity of changing the deposition scheme timescales of deposition into the soil are calculated for one year integrations after injection of small $H_2$ perturbations $r' \ll r_{ref}$, noting that lifetimes will become more homogeneous for longer integrations as the remaining perturbation becomes more mixed into the atmosphere. For sufficiently small perturbations, we assume the chemistry fluxes are unchanged, and the anomalous flux is dominated by the soil flux (see Prather and Holmes, 2013). Likewise, we assume the differences between experiments with the prototype deposition scheme and BF2 are not obscured by using the same chemistry fluxes.

From the soil deposition timescales, an approximate total lifetime, $\tau_{total}$, is calculated as the parallel addition of the lifetime due to chemical loss in the atmosphere $\tau_{atmchem} = 7.7$ yr from Sand et al. (2023), and soil deposition timescales for each perturbation, with an approximate scaling factor $1000/850$ because the 2D model only extends to 150 hPa rather than the top of atmosphere, assuming that $H_2$ mixing ratios in the upper 150 hPa of the atmosphere are similar to those modelled between 1000 and 150 hPa (e.g. Warwick et al., 2004),

$$\tau_{total}^{-1} \approx \tau_{atmchem}^{-1} + \left( \frac{1000 \text{ hPa}}{850 \text{ hPa}} \tau_{soil}^{\dagger,*} \right)^{-1}, \tag{13}$$

where $\tau_{soil}^{\dagger,*}$ are model timescales calculated in different experiments.

In two perturbation experiments, perturbations to $H_2$ are: † as +1 ppb throughout the simulated atmosphere initiated for each month (Fig. 10a); and * as +0.1 Tg in the lower 200 hPa layer through a continuum of latitude bands, and initiated at 0, 90, 180 and 270 days to sample the sensitivity of the soil lifetime to the season of the emission (Fig. 10b). A soil lifetime is then calculated from the decay of each perturbation after one year.

In the prototype scheme, the SH deposition peaks at 10-20°S around 1 month into the year (Figs. 5d and 7b). Whereas, shown in Fig. 9c, the best-fit SH deposition occurs in extended periods spreading northwards across the equator from March to October and in the SH subtropics into the winter. In the NH mid-latitudes, BF2 has a similar temporal signal as the prototype scheme, but with extensive deposition continuing into September.

In simulations using the prototype scheme, a whole-troposphere $H_2$ perturbation has the longest soil lifetime, $\tau_{soil}$, when initialised around 150 days into the year – as a product of the seasonal high mixing ratio in the NH subtropics observed in Fig. 4b. $\tau_{soil}$ has a minimum for perturbations initialised around 30 days, when there is a high rate of deposition in the SH, but is half a year out of phase with the peak deposition, which is concentrated in the NH (Fig. 7a).

Under BF2, $\tau_{soil}$ is decreased relative to the prototype scheme for perturbations injected around either solstice reflecting the stronger seasonality in both hemispheres (resolving Fig. 4a). This decreased lifetime is balanced by relatively longer soil lifetimes at the equinoxes, where perturbations mix and react in the atmosphere over a longer timescale before deposition.

     There is a gradient of longer $\tau_{soil}$ for perturbations initialised near the surface in the SH to shorter $\tau_{soil}$ for those initialised in the NH (Fig. 10b) reflecting the greater soil sink in the NH. For perturbations initialised at low-latitudes, the seasonality

of $\tau_{soil}$ corresponds with those of whole troposphere perturbations. However, this is inverted for perturbations initialised in the extra-tropical NH where $\tau_{soil}$ is longest for perturbations initialised in the autumn and winter. The longer soil deposition timescale in the SH corresponds with a result of Derwent (2023), who identified a monotonically decreasing GWP for $H_2$ emissions sources from the SH mid-latitudes to the NH mid-latitudes.

     The same pattern is largely reproduced for simulations using BF2, but with a weaker meridional gradient. Quadratic fits

of the annual mean $\tau_{soil}$ against equal area intervals (sin of the perturbation latitude, $\lambda$) result in $\tau_{soil} = 2.9 \pm 0.2$ yr at the equator, with a similar first degree gradient of $-1.1 \pm 0.1$ yr with BF2 compared to $-1.2 \pm 0.3$ yr with the prototype scheme, and an insignificant change in this negative gradient with latitude reflecting that BF2 only aims to resolve the seasonality in the observed $H_2$ distribution. However, there are distinct changes in seasonal lifetimes under BF2. In particular: SH extra-tropical perturbations are deposited more quickly when injected at 0 days; and NH extra-tropical perturbations are more slowly

deposited when injected at 90 days.

## 8    Conclusions

The methods we have discussed provide a toolbox to constrain the development of $H_2$ deposition schemes using empirical observations of the zonal mean $H_2$ distribution and seasonality. In particular, we have identified the asymmetry in the seasonal cycle of $H_2$ in the NH and SH. Without a seasonally varying soil uptake, the seasonality of zonal mean surface $H_2$ would

be dominated by the seasonality of atmospheric $H_2$ production and oxidation. $H_2$ concentrations would peak with a similar amplitude during the late-summer to early-autumn in both the NH and SH extra-tropical regions (Fig. 4; the seasonality of the deposition induces a stronger amplitude and earlier peak in the NH $H_2$ signal.

     We have shown that a prototype deposition scheme based on the assumed leading physical-biological processes of soil $H_2$ uptake (Fig. 5d) is able to effectively capture some key features of the planetary $H_2$ distribution in the 2D model. Assuming the

365 annual-mean of this prototype scheme as a suitable basic deposition state, we then produced a 'best-fit' deposition scheme that reproduces the planetary $H_2$ seasonality independent of the seasonality of the prototype scheme. Comparing the seasonality of the best-fit scheme against the prototype scheme tested the accuracy of the seasonality of the prototype scheme. This

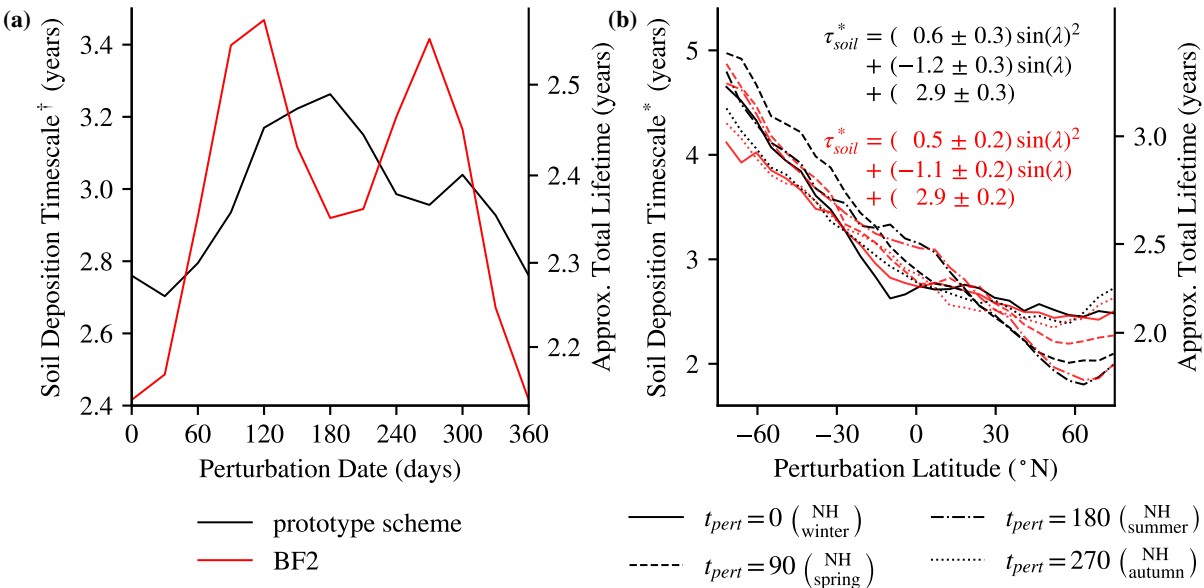

**Figure 10.** Soil timescales calculated from 1 year simulations with: (**a**) $\tau^{\dagger}_{soil}$ calculated with whole troposphere 1 ppb perturbations initialised at a range of dates; and (**b**) $\tau^{*}_{soil}$ calculated with perturbations of 0.1 Tg in the lower 200 hPa in latitude bands – quadratic fits are made against sin latitude. In both cases the soil lifetime is underestimated as the atmosphere is only simulated though 1000-150 hPa and due to the relatively short simulation length; and in $*$ near surface perturbation configuration. Approximate $\tau_{total}$ is the harmonic sum of $\frac{1}{0.85}\tau^{\dagger,*}_{soil}$ and $\tau_{atmchem}$, which is assumed to have the constant value 7.7 yr from Sand et al. (2023). Likewise, the 3.4 yr soil deposition lifetime (from Sand et al., 2023) corresponds with 2.9 yr in this scheme.

challenges the assumed deposition seasonality in the tropics, providing useful insight for where similar deposition schemes should be revised to improve the accuracy in future $H_2$ modeling efforts.

In the NH extra-tropics the prototype scheme performs well at reproducing the observed annual mean meridional gradient and seasonality of surface $H_2$ mixing ratios. Simulations in the toolbox model produce too-high $H_2$ mixing ratios in the SH, where Paulot et al. (2024) have reduced SH net ocean emissions in the extra-tropics compared with emissions used in this study. However, we show how differences in phase and a too-weak amplitude of seasonality in the Southern tropics and subtropics may be resolved by the choice of the deposition scheme. We find that while the prototype scheme agrees with key features

of the best-fit deposition scheme, the prototype deposition scheme would better reproduce observations with a lag of +2 to +3 months in its seasonality in the tropics. Other sources of hysteresis on the seasonality may be explained by dependence on: irreversible degeneration of free enzymes in soils where seasonal temperatures fluctuate above 30°C (Chowdhury and Conrad, 2010); variations in the soil organic carbon content (King et al., 2008; Karbin et al., 2024); or even the life-cycle of soil microbes (Meredith et al., 2014). Our results indicate that deeper investigations into the $H_2$ flux in tropical soils are needed

to build our understanding of the links between these soil microbial processes and the planetary scale $H_2$ signal.

A second best-fit deposition scheme was found by perturbing the monthly-varying prototype scheme (Fig. 9c). Both in the prototype and best-fit deposition schemes there is both a strong seasonal and meridional dependence of the soil deposition timescale for different configurations of $H_2$ perturbations. The choice of the deposition scheme also had an impact on the meridional dependence of the soil lifetime for $H_2$ emissions injected at particular times of year.

When calculating the climate benefit of future hydrogen energy systems, such as by Hauglustaine et al. (2022), constraining the $H_2$ flux to soils may have a significant impact for the sensitive question of how accurately comprehensive models predict the spatial dependence of environmental impacts of $H_2$ emissions from regional industrial hydrogen projects (e.g. Derwent, 2023). This is particularly important in the SH, where soil deposition timescales are shorter for emissions during the summer in simulations with the best-fit scheme compared with the prototype scheme.

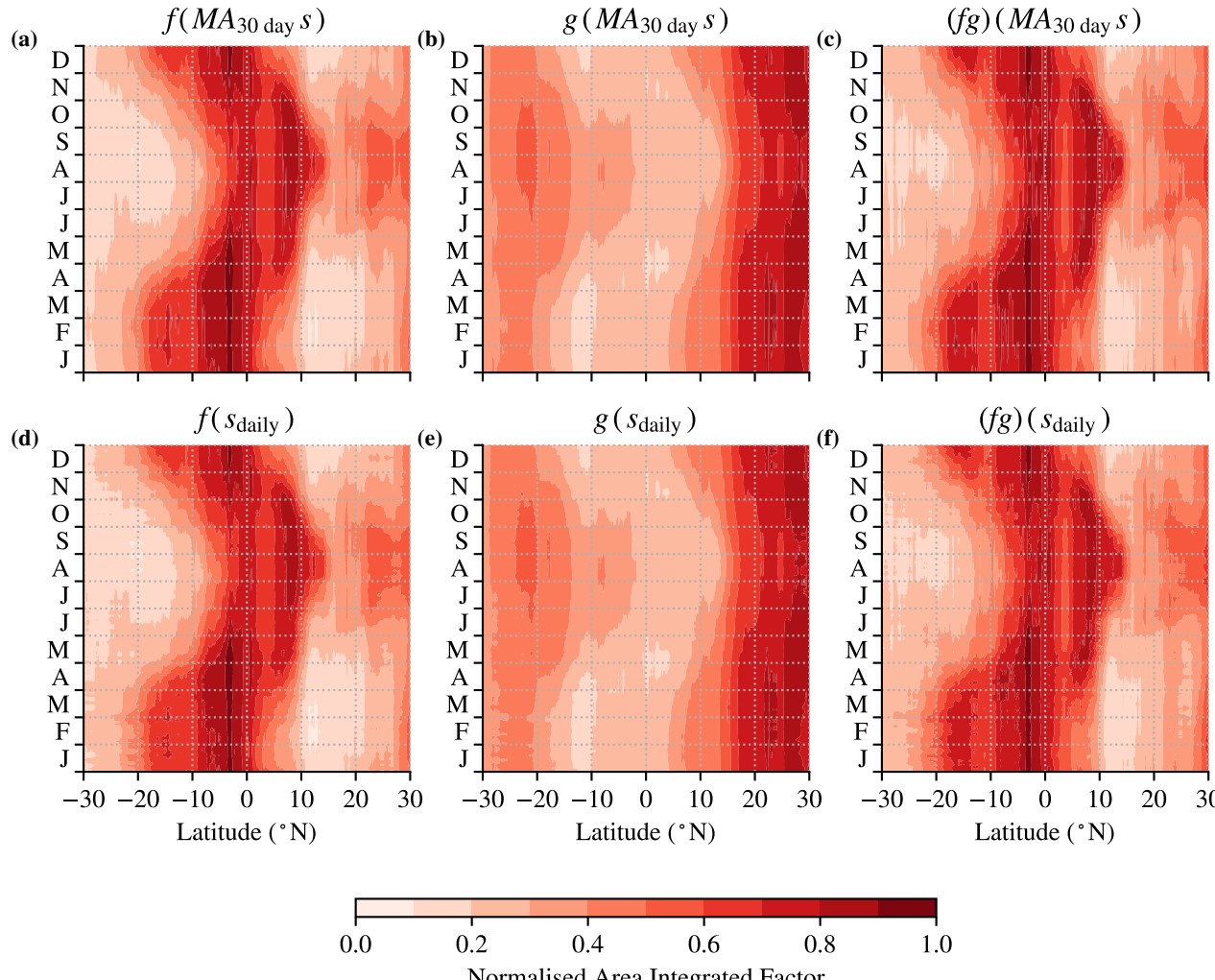

**Figure A1.** Comparison of soil moisture dependent factors in the prototype deposition scheme under different temporal resolution ERA5 2010-2020 data in $30°$S-$30°$N. (**a**) and (**d**) compare $f$; (**b**) and (**e**) compare $g$; and (**c**) and (**f**) compare the product $fg$. (**a**)-(**c**) are calculated with 30-day moving average soil moisture, $MA_{30\,day}s$; and (**d**)-(**f**) are calculated with daily average soil moisture. The data are averaged for day in year excluding leap days. Area integrated factors are normalised with the same multiplier in: (**a**) and (**d**); (**b**) and (**e**); and (**c**) and (**f**).

## Appendix B

Anomalous sites are determined based on RMS error of the sum of the first and second harmonics, $h_1 + h_2$, versus the mid-filtered station time-series, $F_{mid}(\text{data})$;

$$\text{anomalous:} \sqrt{\sum_t \left(h_1 + h_2 - F_{mid}(\text{data})\right)^2} > 20 \text{ ppb.} \tag{B1}$$

This criterion identifies five station timeseries where the decomposition method could not fit harmonics (illustrated in Fig. 3).

## Appendix C

Table C1: Station data plotted in Figs.2 and 4 with codes from Pétron et al. (2023) and difference versus best-fit average mixing ratio. Local prototype deposition is the average deposition within $\pm 2.5°$N and $\pm 2.5°$E: no deposition; low deposition $\in (0, 0.2]$ g m$^{-2}$ yr$^{-1}$; and high deposition $> 0.2$ g m$^{-2}$ yr$^{-1}$. RMSE$(h_1 + h_2)$ is the RMS error of the fit versus the data (Equation B1).

| Code | Latitude (°N) | Longitude (°E) | 2012-2018 avg. mixing ratio (ppb) | Difference vs. best-fit (ppb) | A (ppb) | Φ (days) | Local prototype deposition | RMSE $(h_1+h_2)$ (ppb) | Anomalous |
|------|------|------|------|------|------|------|------|------|------|
| SPO | -90.0 | -24.8 | 548.9 | -0.2 | 9.4 | 54 | low | 1 | |
| HBA | -75.5 | -25.6 | 551.5 | 0.8 | 8.0 | 56 | low | 3 | |
| SYO | -69.0 | 39.6 | 549.6 | 0.1 | 10.0 | 53 | low | 3 | |
| PSA | -64.8 | -64.1 | 547.9 | -0.7 | 9.3 | 54 | low | 1 | |
| USH | -54.8 | -68.3 | 547.9 | 0.3 | 14.3 | 51 | low | 8 | |
| CRZ | -46.4 | 51.8 | 545.9 | -1.4 | 10.6 | 47 | none | 5 | |
| BHD | -41.4 | 174.9 | 548.6 | 0.4 | 11.8 | 31 | low | 6 | |
| CGO | -40.7 | 144.7 | 549.8 | 1.5 | 19.7 | 38 | low | 14 | |
| CPT | -34.4 | 18.5 | 574.5 | 26.2 | 22.4 | 31 | low | 12 | |
| EIC | -27.2 | -109.4 | 544.2 | -2.7 | 69.7 | 363 | none | 52 | ○ |
| NMB | -23.6 | 15.0 | 546.0 | -1.5 | 15.3 | 2 | low | 7 | |
| SMO | -14.2 | -170.6 | 554.9 | 2.4 | 13.4 | 30 | none | 7 | |
| ASC | -8.0 | -14.4 | 551.6 | -2.4 | 8.8 | 342 | none | 4 | |
| NAT | -5.8 | -35.2 | 556.6 | 2.3 | 13.7 | 49 | low | 11 | |
| SEY | -4.7 | 55.5 | 558.4 | 4.1 | 19.7 | 22 | none | 11 | |
| BKT | -0.2 | 100.3 | 563.0 | 9.6 | 26.4 | 331 | low | 24 | ○ |
| CHR | 1.7 | -157.2 | 551.8 | -0.9 | 15.0 | 74 | none | 11 | |

| | | | | | | | | | |
|---|---|---|---|---|---|---|---|---|---|
| RPB | 13.2 | -59.4 | 547.5 | -1.8 | 11.6 | 172 | low | 5 | |
| GMI | 13.4 | 144.7 | 552.5 | 3.2 | 37.0 | 158 | none | 19 | |
| MEX | 19.0 | -97.3 | 547.9 | -0.9 | 17.6 | 129 | high | 10 | |
| MLO | 19.5 | -155.6 | 540.2 | -8.5 | 13.7 | 143 | none | 6 | |
| DSI | 20.7 | 116.7 | 558.1 | 9.7 | 29.4 | 118 | low | 15 | |
| ASK | 23.3 | 5.6 | 548.0 | 0.9 | 20.1 | 134 | none | 16 | |
| LLN | 23.5 | 120.9 | 557.1 | 10.1 | 23.5 | 119 | low | 13 | |
| KEY | 25.7 | -80.2 | 547.9 | 2.9 | 24.5 | 161 | low | 11 | |
| IZO | 28.3 | -16.5 | 542.3 | 1.0 | 15.2 | 173 | none | 5 | |
| BMW | 32.3 | -64.9 | 536.8 | 3.0 | 22.8 | 171 | none | 8 | |
| WLG | 36.3 | 100.9 | 524.8 | 0.3 | 24.3 | 117 | high | 14 | |
| AMY | 36.5 | 126.3 | 549.7 | 25.7 | 38.6 | 144 | high | 19 | |
| TAP | 36.7 | 126.1 | 514.5 | -9.1 | 42.1 | 159 | high | 16 | |
| AZR | 38.8 | -27.4 | 519.2 | -0.3 | 16.0 | 147 | none | 7 | |
| UTA | 39.9 | -113.7 | 504.8 | -12.8 | 51.0 | 176 | high | 18 | |
| SDZ | 40.6 | 117.1 | 565.5 | 48.8 | 111.8 | 181 | high | 60 | ○ |
| THD | 41.1 | -124.2 | 524.8 | 8.6 | 28.1 | 141 | high | 11 | |
| CIB | 41.8 | -4.9 | 504.9 | -10.5 | 30.9 | 159 | high | 13 | |
| UUM | 44.5 | 111.1 | 495.1 | -19.0 | 47.2 | 116 | high | 23 | ○ |
| LEF | 45.9 | -90.3 | 504.3 | -9.7 | 27.8 | 118 | high | 7 | |
| HUN | 47.0 | 16.7 | 527.9 | 13.9 | 27.0 | 122 | high | 12 | |
| HPB | 47.8 | 11.0 | 523.0 | 9.0 | 21.2 | 128 | high | 7 | |
| OXK | 50.0 | 11.8 | 521.9 | 8.3 | 16.1 | 128 | high | 7 | |
| SHM | 52.7 | 174.1 | 510.1 | -2.1 | 28.4 | 121 | none | 10 | |
| MHD | 53.3 | -9.9 | 519.5 | 7.7 | 18.8 | 120 | low | 7 | |
| LLB | 55.0 | -112.5 | 731.0 | 220.4 | 303.8 | 44 | high | 189 | ○ |
| CBA | 55.2 | -162.7 | 501.9 | -8.6 | 26.6 | 115 | none | 9 | |
| ICE | 63.4 | -20.3 | 512.5 | 3.5 | 24.6 | 106 | low | 9 | |
| BRW | 71.3 | -156.6 | 503.1 | -4.9 | 28.7 | 121 | high | 8 | |
| TIK | 71.6 | 128.9 | 489.9 | -18.0 | 39.4 | 117 | low | 10 | |
| SUM | 72.6 | -38.4 | 520.3 | 12.5 | 19.4 | 131 | low | 4 | |
| ZEP | 78.9 | 11.9 | 507.2 | 3.9 | 23.6 | 106 | low | 8 | |
| ALT | 82.5 | -62.5 | 493.5 | -5.3 | 33.5 | 90 | low | 14 | |

**Appendix D**

To analyse how adjustments to the seasonality of the prototype deposition scheme affect results we define the ratio of the RMS error between the seasonality of an adjusted deposition scheme and BF1 to the RMS seasonality of BF1 at each latitude:

$$R_{BF}(\alpha, \Delta t) = \frac{\sqrt{\int_t^{t+1 \text{ year}} \left( r_{ref}^{-1}\text{BF1} - \alpha W'(t' - \Delta t) \right)^2 dt'}}{\sqrt{\int_t^{t+1 \text{ year}} \left( r_{ref}^{-1}\text{BF1} \right)^2 dt'}}, \tag{D1}$$

where the adjusted deposition seasonality $\alpha W'(t - \Delta t)$ is the seasonality of the prototype scheme, $(fgh)'$, scaled by a factor $\alpha$ and offset in time by $\Delta t$.

*Author contributions.* ATC – ideation, analysis design, performing analysis, writing; and DS – ideation, analysis design, writing.

*Competing interests.* No competing interests are present.

*Acknowledgements.* This work was funded at The University of Edinburgh as part of the HECTER project under NERC grant number (NE/X012735/1). Hannah Bryant for UKCA chemistry outputs from earlier work (Sand et al., 2023); Fabien Paulot for estimated emissions files from earlier work (Paulot et al., 2021); those aforesaid and Richard Derwent, Saeed Karbin, Keith Shine and Joanne Smith for helpful and motivating discussions. This work used JASMIN, the UK's collaborative data analysis environment (https://www.jasmin.ac.uk ).

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
