# Peer review of "Soil Deposition of Atmospheric Hydrogen Constrained using Planetary Scale Observations"

_EGUsphere, 2024_

## Community Comment (CC1)

$f$ (soil moisture, soil texture)  $g_s$ (soil moisture, soil texture)  $h$ (soil temperature)

Normalised Factor

Soil Moisture Content   Soil Moisture Content   Soil Temperature (°C)

- - coarse   - - medium fine   - - very fine
- - medium   - - fine   - - (tropical) organic

---

## Editor Comment (EC1)

**Review of MS entitled: "Soil Deposition of Atmospheric Hydrogen Constrained using Planetary Scale Observations" by Chaudhri and Stevenson**

**General Comments:**

This manuscript presents a novel method to optimize atmospheric $H_2$ deposition modeling using zonal-mean seasonality from the NOAA $H_2$ dataset. By applying high- and low-pass filtering to perturb a prototype deposition scheme based on soil temperature and moisture dynamics, a "best-fit" scheme is identified that more accurately reproduces observed $H_2$ patterns. The findings reveal a necessary +3-month shift in microbial consumption seasonality in the tropics, underscoring the need for improved $H_2$ flux constraints in low-latitude regions.

While the proposed observation-based methodology offers valuable insights into $H_2$ deposition, several critical concerns must be addressed to ensure the robustness of the findings.

First, the sensitivity of the "best-fit" estimates to assumptions about the spatial and temporal distribution of the OH sink remains unclear. Given that OH chemistry is derived from a single model with prescribed $H_2$ and $CH_4$ concentrations, how can the deposition sink be effectively disentangled from the chemical sink? Could variations and uncertainties in OH fields also influence the zonal-mean seasonality? Despite $H_2$'s relatively long lifetime with respect to OH, the spatial and temporal heterogeneity of OH should be considered interactively.

Second, beyond microbial uptake and soil diffusion, could other processes influence deposition? For instance, could atmospheric turbulence and its variability across latitudes contribute to seasonal variations?

Third, how does the 2D model compare to seasonal patterns from 3D simulations? Would leveraging 3D model outputs (e.g., Sands et al., 2023) provide a more appropriate framework for optimizing deposition?

Fourth, strengthening the manuscript's impact would require presenting site-specific seasonality for all measurement sites, not just Mace Head. This would better highlight regional differences and their connection to zonal-mean anomalies.

Finally, for the method to be reproducible, it would be best to elaborate on the rationale behind specific assumptions and thresholds, as well as describe the uncertainties on derived quantities of the filtering algorithm (e.g., phase and amplitude).

**Specific Comments:**

Line 55: "we assume that the general effect of these fluxes is well approximated when they are modelled in their zonal-mean monthly-mean". How valid is this assumption? Can you please elaborate.

Line 71: Can you please elaborate on the 'spatial filtering' used?

Figure 1 caption (also Line 95-97): What is the rationale behind choosing 20 ppb for the RMS error threshold for it to be an anomaly? Same for the Gaussian filter of $\sigma=5$ degrees latitude.

Line 78-84: How sensitive are your findings to assumptions on timescales (<30 days and 1 year) for these filters? How robust is this method if other filtering methods are applied (e.g., singular spectrum analysis – SSA, Seasonal Trend decomposition using Loess – STL)?

Line 101: "The seasonal $H_2$ signal does not depend significantly on zonal variations in local deposition." Why do you think this is?

Line 106-108: "This supports the assumption that the deposition into soils is dominated by the larger land area of the NH, where this soil sink exceeds anthropogenic emissions and the net source of $H_2$ from atmospheric chemistry (Paulot et al., 2021)." Can this be attributed as well to variations in OH in NH, tropics, and SH?

Line 129-140: How large is the influence of diffusion into soil on deposition relative to microbial activity? Aren't these processes coupled in reality?

Line 155-160: While the 2D model is justified, what would be the sensitivity to the best fit when 3D models are used?

Line 176-180: What is the physical rationale behind the choice of assuming that fluxes are independent to $H_2$ mixing ratio?

Line 197-200: Interesting finding regarding the cross-term. Why do you think this is the case?

Line 217: How sensitive would the best fit to assumptions/representations of P and rD, and M?

---

## Author Response (AR1)

Dear the referees and the editor, Jens-Uwe Grooß, and for the interest of the community discussion,

Thank you all for your insightful and helpful comments and recommendations to improve our manuscript. In the revised manuscript we have included diffusivity in the prototype deposition scheme. This change required re-running each simulation in the paper and there have been changes to figures and text throughout to reflect this change. In general, as RC2 suggested, diffusivity has the strongest effect effect in moist soils – particularly in the tropics. Additionally, we have made many text and figure changes to improve readability and provide more clarity for our results and analysis.

Please find detailed responses to individual reviewer comments below.

Yours sincerely and with appreciation,

Alex Tardito Chaudhri and David Stevenson

Responses to RC1:

*1) Figure 1 shows the sites with different deposition rates, anomalous means, and late peaks. Can you please identify these stations so the reader can cross-compare with the NOAA dataset? This information can be provided in Supplementary Materials. There may be some local factors that can explain the statistically anomalous sites. For example, there may be local pollution sources, strong local soil sinks, variable sampling times, or potential sampling issues. Sites identified by your work might be tagged for further scrutiny. It would also be helpful to describe what is meant by 'low' vs 'high' deposition, and how this was assessed for each site.*

Thank you for this request to help the reproducibility and broaden the application of this work. We have included an extensive table in the appendix section recording each of these values. The low versus high deposition was simply a way of coding the stations against the prototype scheme based on a local 0.2 g/m^2/yr deposition criterion. This is explained both in the caption and in the appendix table.

*2) Line 43. The Meredith et al., 2017 paper, appears to be the appropriate reference here, in lieu of their 2014 paper.*

Thank you for this recommendation, yes that is a more appropriate reference.

*3) Line 101: "The seasonal H₂ signal does not depend significantly on zonal variations in local deposition". Can you please clarify what is meant by this sentence? What does it mean for local deposition to vary across latitudes?*

To make this clearer the sentence has been changed for more explanation:

(ln 154) The small spread of the phase of the seasonality of \chem{H_2} observations from the best-fit shows that the mid-filtered signal varies more with the latitude of sites than due to zonal variations in local deposition.

*4) Line 153-154: In the model, the "long lifetime of H₂ compared with timescales of horizontal mixing in the atmosphere indicating that H₂ is reasonably well mixed across zonal bands". Is this true for all latitudes, including the tropics which have a much larger circumference (and hence mixing time)? There can also be large variability within single latitudinal bands even without identified strong local deposition. (e.g., it appears there can be 30 ppb variation in the mean at specific latitudes, even those that have not been identified as anomalous).*

Thank you for pointing out the problems with this assumption. We have added some explanation that at certain latitudes there is observed zonal variation c.5% of the annual mean:

(ln 189) Additionally, we note the long average lifetime of H_2 compared with timescales of horizontal mixing in the atmosphere (e.g. {pierrehumbert1993}) indicating that H_2 is reasonably well mixed across zonal bands. However, this assumption is challenged at latitudes with particularly strong sources and sinks. For example, there is a spread of c.5% in non-anomalous annual mean mixing ratios around 40N (Fig. 1b).

*5) "Line 224-226: "In particular, the seasonality due to soil temperature captures the strong seasonality of BF1 in the NH mid-latitudes, which has been identified in microbial laboratory studies (Smith-Downey, 2006)." This sentence may need to be reworded. The microbial laboratory studies reveal the temperature sensitivity of H₂ deposition, but do not themselves identify the strong seasonality in NH mid-latitudes.*

Hopefully this is more accurate in 2 sentences: one on capturing BF1; one on mid-latitude seasons and soil temperature sensitivity. Becomes:

(ln 262) The seasonality of the prototype scheme, (fgh)', reproduces some key features of BF1. In particular, the prototype scheme captures the strong seasonality of BF1 in the NH mid-latitudes. There, seasonality is driven by the sensitivity of microbial activity to variation in soil temperature, which has been identified in laboratory microbial studies {smith-downey2006}.

*6) Line 311-313. "Without a seasonally varying soil uptake, zonal mean surface H₂ would peak with a similar amplitude during the late summer to early autumn in both the NH and SH extra-tropical regions". Is this based primarily on estimated emissions, oxidation and mixing? What are the most important factors strongly influencing the modeled seasonality when soils are not included?*

Rewritten to identify the inter-hemispheric symmetry in the atmospheric production and destruction:

(ln 368) Without a seasonally varying soil uptake, the seasonality of zonal mean surface $H_2$ would be dominated by the seasonality of atmospheric $H_2$ production and oxidation. $H_2$ concentrations would peak with a similar amplitude during the late-summer to early-autumn in both the NH and SH extra-tropical regions (Fig. 3); the seasonality of the deposition induces a stronger amplitude and earlier peak in the NH $H_2$ signal.

*7) [optional] Lines 235-237. It would help the reader if the specific locations discussed are visually identified in figures 8a and 8b, such as with a circle, arrow or by indicating "areas of darker blue shading".*

Thank you for this helpful suggestion. Clear arrows (Fig. 8) have been added to show where the adjustments are that are discussed in the text. Also, with the updated prototype deposition scheme there are only 3 latitudes with peaks in seasonality – this makes the discussion somewhat clearer too (lns 268-290).

*8) Lines 240-243 "In the NH, better agreement is achieved where this lag and the amplitude of the seasonality of the peaks are decreased with increasing latitude; the peak at 52N (D) agrees well with BF1 for a ~1 week lag and a 60% multiplier". This statements suggests a steady decrease in the amplitude and lag time as you go towards the poles. Is this true, or does it refer to a specific high latitude band?*

Thank you for pointing this out. Originally the discussion referred just to the peak amplitude latitudes. This has been extensively and carefully re-written to make sure the interpretation is clear:

In Fig. 8c the latitudes of peak amplitude of seasonality in the prototype scheme are isolated for optimised agreement with BF1 under adjustments to the seasonality multiplier and offset. In both deep tropical peaks (A,B), better agreement occurs for a lag of two to three months. Additionally, the strongest agreement occurs for a weaker seasonality in the SH tropical peak (A). In the NH mid-latitudes better agreement is achieved with a one week lag in the peak deposition at 52∘N (C).

Figure 8c shows that RBF is minimised for decreases in the amplitude of the seasonality at A, B, C. However, this arises as BF1 is unconstrained by the seasonality in the prototype scheme, instead assuming variations in deposition spread across latitudes. By integrating the deposition seasonality across wider latitude bands, Figs. 8d,e show a lag and increases in amplitude of seasonality between 30∘S and 30∘N; whereas Fig. 8f shows differences in the seasonal signal in the NH mid-latitudes, but with a similar amplitude (cf. Figs.7a,b).

*9) Lines 264-265: "enhanced deposition with a maximum of around 0.2 Tg yr-1/ lat focused in the NH mid-latitudes, and peaking in both hemispheres between December and February (Fig 9b)". A Dec-Feb peak in deposition is not obvious in Figure 9b.*

Thank you for pointing out that that was not clear. This figure has changed slightly in the new version due to the new simulations. The figure has been replotted with different

contour intervals that make these peaks clearer to see. The same intervals are now used in Figs. 5,7,9.

*10) Lines 322-323. "In the SH the prototype scheme results in too-high surface mixing ratios in the annual mean, and differences in phase and a weak amplitude of the seasonality in the Southern tropics and subtropics". Are your results consistent with a small but widespread additional sink in the Southern Hemisphere, such as a weak ocean sink? This may be too speculative for the manuscript, but it may be helpful to question the role of the oceans in the $H_2$ budget.*

This has been changed to reference Paulot et al. 2024 who revise SH ocean emissions (included in their supplementary material).

(ln 380) Simulations in the toolbox model produce too-high H2 mixing ratios in the SH, where Paulot et al. (2024) have reduced SH net ocean emissions in the extra-tropics compared with emissions used in this study. However, we show how differences in phase and a too-weak amplitude of seasonality in the Southern tropics and subtropics may be resolved by the choice of the deposition scheme.

Responses to RC2:

*I suggest the authors better articulate this work's distinctions and advancements compared to Paulot [Paulot et al. (2024, ACP)]*

In the introduction we provide a clearer link for how this work builds on and complements that excellent study:

(ln 52) Our results and analysis also complement Paulot et al. (2024), who have revised H2 emissions and deposition estimates to better-reproduce the H2 distribution and seasonality of clustered observations. They identified major gaps in our understanding of soil removal of H2. We analyse the soil uptake arising from a model constrained to reproduce a latitude varying H2 seasonality, while relaxing assumptions on the estimated soil uptake.

In the analysis of observations, we highlight a key distinction:

(ln 156) We exploit a $H_2$ seasonality that continuously varies with latitude as a distinct constraint compared to Paulot et al. (2024). This constraint allows us to test assumptions in the prototype deposition scheme at higher meridional resolution (Sec. 5).

In the conclusion we also refer to their revised ocean emissions.

*By neglecting diffusion, the authors exclude a critical physical factor that substantially impacts soil sink representation. This omission introduces large biases in calculated soil uptake rates, particularly in humid temperate and tropical regions with more pronounced diffusion limitations. As a result, both the prototype scheme and the subsequent best-fit scheme are fundamentally flawed, rendering the conclusions unreliable. Addressing this issue requires incorporating diffusion into the soil sink parameterization and rerunning the entire modeling framework (prototype and best fit).*

Thank you for identifying this important point. Further to my (ATC) earlier reply in the community discussion, I have implemented a representation of diffusion limiting uptake into the prototype deposition scheme. This required a rewrite of former Sec. 3 (now Sec. 2 on the prototype scheme) that is too long to include here. As you suggested, the change to this new prototype scheme had an impact on uptake in moist soils (mostly tropical).

After re-running all simulations and analysis, the use of the new prototype scheme required changes to all figures (except the example decomposition at Mace Head). In Fig. 6, we show how strongly this inclusion of diffusion contributes to seasonality. Text changes reflect the changes. However, while BF1 and BF2 are changed a small amount, the new deposition scheme was still unable to produce the SH Feb-Apr and NH Aug-Oct peak uptake in the tropics in each hemisphere required to reproduce the best-fit observed seasonality.

Additionally, in the appendix we have included a comparison of soil moisture dependent uptake terms in the prototype scheme under 30-day rolling average and daily average inputs. This is mentioned in the text:

A comparison of f and g with soil moisture varying on 30-day and daily timescales is provided in Fig. A1. The use of monthly average ERA5 data can lead to slightly faster uptake rates in semi-arid regions (cf. Bertagni et al., 2021) but does not substantially affect the seasonality.

*In addition to this central issue, the clarity and structure of the manuscript could be significantly improved. Just as examples: i) the results in terms of atmospheric concentrations are presented before introducing the underlying soil sink calculations, which disrupts the logical flow of the paper; ii) section 5 on the best-fit scheme includes mathematical equations that appear simultaneously overly simplistic and excessively detailed for inclusion in the main text (appendix?); iii) the abstract could be simplified to be less technical.*

Thank you for these suggestions regarding clarity and structure. We have moved the equations showing the test for anomalous station time-series and ratio of RMS error in seasonality versus BF1 to the RMS BF1 seasonality to the appendix.

We have made text changes throughout to boost clarity and readability.

In the abstract we have identified specific months that we are comparing in order to make our fundings easier to understand.

Responses to RC3:

*While the proposed observation-based methodology offers valuable insights into H₂ deposition, several critical concerns must be addressed to ensure the robustness of the findings. First, the sensitivity of the "best-fit" estimates to assumptions about the spatial and temporal distribution of the OH sink remains unclear. Given that OH chemistry is derived from a single model with prescribed H₂ and CH₄ concentrations, how can the deposition sink be effectively disentangled from the chemical sink? Could variations and uncertainties in OH fields also influence the zonal-mean seasonality? Despite H₂'s relatively long lifetime with respect to OH, the spatial and temporal heterogeneity of OH should be considered interactively. Second, beyond microbial uptake and soil diffusion, could other processes influence deposition? For instance, could atmospheric turbulence and its variability across latitudes contribute to seasonal variations? Third, how does the 2D model compare to seasonal patterns from 3D simulations? Would leveraging 3D model outputs (e.g., Sands et al., 2023) provide a more appropriate framework for optimizing deposition? Fourth, strengthening the manuscript's impact would require presenting site-specific seasonality for all measurement sites, not just Mace Head. This would better highlight regional differences and their connection to zonal-mean anomalies. Finally, for the method to be reproducible, it would be best to elaborate on the rationale behind specific assumptions and thresholds, as well as describe the uncertainties on derived quantities of the filtering algorithm (e.g., phase and amplitude).*

Thank you for these insightful and helpful questions, we have integrated many of these into the textual changes in the revised manuscript. We have attempted to answer these in the responses to individual comments (below) or in responses to R1 and R2 (above).

*Line 55: "we assume that the general effect of these fluxes is well approximated when they are modelled in their zonal-mean monthly-mean". How valid is this assumption? Can you please elaborate.*

This is an important point, and we have extended the discussion of these assumptions in the model formulation section:

e.g. (ln 190) However, this assumption is challenged at latitudes with particularly strong sources and sinks. For example, there is a spread of c.5% in the non-anomalous observed annual mean mixing ratios around 40◦N (Fig. 2b).

*Line 71: Can you please elaborate on the 'spatial filtering' used?*

Thank you for this request, we have included more explanation:

(ln 174) The width of the filter was chosen to produce a smooth best-fit between observations that preserves the distinct patterns with latitude that vary over scales c.10◦N.

*Figure 1 caption (also Line 95-97): What is the rationale behind choosing 20 ppb for the RMS error threshold for it to be an anomaly? Same for the Gaussian filter of σ=5 degrees latitude.*

Further to the previous reply, the criterion to reject anomalous sites isolated timeseries where 1st and 2nd harmonics could not easily be fitted. This was checked with inspection. Generally, these sites suffered inconsistencies in the data not seen in the other sites, but we do not attempt a deep discussion or speculation as to why this might be the case. In the appendix we have included an extended discussion of the equation used to identify these and shown the RMS error in a table for each site.

(ln 148) Anomalous results are categorised where this RMS error is greater than 20 ppb (Equation B1, sites indicated in Fig.2). By inspection, this criterion effectively excludes five sites where harmonics could not be identified (such as in Fig. 3).

*Line 78-84: How sensitive are your findings to assumptions on timescales (<30 days and 1 year) for these filters? How robust is this method if other filtering methods are applied (e.g., singular spectrum analysis – SSA, Seasonal Trend decomposition using Loess – STL)?*

This is a very interesting question and would be relevant for future research that could attempt to decode the seasonal signal at a finer resolution, or to infer regional fluxes by relating wind-direction to the sub-seasonal noise. However, the data are rather limited due to their sampling frequency, typically every one or two weeks (ln 120), and differences in sampling conditions at different sites, so analysis on shorter timescales is complicated. This being said, the data are generally strongly dominated by the change on the annual timescale, with strong consistency over the 2012-2018 time period (where these data existed), so this harmonic is particularly useful for this analysis.

*Line 101: "The seasonal $H_2$ signal does not depend significantly on zonal variations in local deposition." Why do you think this is?*

We justify this as zonal mixing timescales are much shorter than the seasonal timescale, the bulk mass of $H_2$ divided by emission and production rates, and the average lifetime of $H_2$ in the atmosphere. However, please see reply above where we identify c.40N as having more variation because of stronger fluxes (more land surface and anthropogenic activity).

*Line 106-108: "This supports the assumption that the deposition into soils is dominated by the larger land area of the NH, where this soil sink exceeds anthropogenic emissions and the net source of $H_2$ from atmospheric chemistry (Paulot et al., 2021)." Can this be attributed as well to variations in OH in NH, tropics, and SH?*

Thank you for this question, we include a simulation where annual mean deposition velocities are used (Fig. 4 -dotted) to demonstrate the seasonality that would occur without seasonally varying uptake. In the revised manuscript, we have improved the clarity of the discussion on this:

The importance of deposition seasonality is indicated in Fig. 4: a simulation with the same annual deposition flux, but without deposition seasonality, (dotted line) fails to reproduce key features of the planetary H2 seasonality. In the simulation with H2 seasonality driven by emissions, deposition and atmospheric chemistry (solid line), in both hemispheres H2 peaks in late-summer to early-autumn – February-March in SH and August-September in NH –

with similar zonal-mean peak amplitude 10-15 ppb to the observations in the mid-latitudes. Seasonally varying deposition is required for NH H2 to peak earlier in the year and to resolve the distinct latitude bands of peak seasonality. Alternately, prototype deposition seasonality has little impact in the SH, due to the relative lack of land. In the SH, the seasonality of H2 is mainly controlled by atmospheric chemistry.

*Line 129-140: How large is the influence of diffusion into soil on deposition relative to microbial activity? Aren't these processes coupled in reality?*

Thank you for pointing this out. In the revised manuscript we have included a representation of diffusion into the prototype deposition scheme and explored the consequences of this in detail. In Figure 6, we now include the coefficient of variation of each of these terms and identify how diffusion (g) varies as fraction of its annual mean with latitude. It is most important in moist tropical soils.

*Line 155-160: While the 2D model is justified, what would be the sensitivity to the best fit when 3D models are used?*

This is a very relevant question. Currently, this is complicated by current poor constraints on deposition and emissions, so very few models have simulated $H_2$ with emissions and interactive deposition (an example would be Paulot et al., 2021) – but such models have tended to struggle or fail to capture the observed seasonality with high and low mixing ratios occurring too early in the year. Instead, models have tended to impose a surface boundary condition that can vary based on observations (such as the UKCA instance – that had been used in Sand et al. 2023 – from which we sourced chemistry fluxes).

*Line 176-180: What is the physical rationale behind the choice of assuming that fluxes are independent to $H_2$ mixing ratio?*

The attempt is to control fluxes except the deposition, where atmospheric production and destruction of $H_2$ are linked to a number of coupled reactions and feedbacks (e.g. Warwick et al., 2023) that would not be suitable to attempt to capture in this toolbox model. Additionally, in each of our experiments, $H_2$ mixing ratios did not deviate substantially (as a fraction of their absolute value) from the observations (related to BCs of input data). By assuming unchanged chemistry fluxes, we were then able to calculate BF1 and BF2 (where $H_2$ was constrained to follow observations) and compare these to the prototype experiment. We have made textual changes to clarify our methods and justifications for different assumptions.

*Line 197-200: Interesting finding regarding the cross-term. Why do you think this is the case?*

This is to do with seasons in tropics – similar temperatures and similar h(T) through the year but changing soil moisture with dry/rainy seasons; versus extra tropics – temperatures vary greatly.

*Line 217: How sensitive would the best fit to assumptions/representations of P and rD, and M?*

This is a very relevant point for future research – particularly as our understanding of $H_2$ production by atmospheric chemistry advances. As the soil uptake is generally considered to account for 70-80% of the $H_2$ uptake, and we find that seasonality is comparable with the annual mean, changes in the atmospheric loss would not have a strong influence on the conclusions of this work. We do provide a discussion of sensitivity to emissions (ln 310). We tune M based on reproducing observed $SF_6$ from an emissions inventory. However, beyond general arguments about changing meridional mixing rates, exploring the role of mixing in depth would require some representation at least of synoptic weather, or in the case of turbulence the role of diurnal variations in boundary layer thickness and mixing may be important to investigate deeply.

---

## Author Response (AR2)

Dear Jens-Uwe Grooß and Referees,

Thank you for reviewing our revised manuscript and offering your support for its publication.

We have made changes following the referee suggestions and included (below) is a line-by-line response to Referee #2.

Yours sincerely,

Alex Tardito Chaudhri and David Stevenson

Responses to Referee #2:

**RC: The authors may wish to discuss whether the mismatches observed in tropical regions are partly driven by uncertainties in input data (in addition to parameter choice and process representation), which can significantly influence modeled seasonal cycles. Paulot et al. (2024), for example, show (Fig. 3a) that the GFDL moisture dataset produces a single harmonic in the tropics, similar to the authors' prototype scheme using ERA5, whereas the GLDAS dataset yields a double harmonic, consistent with the best-fit scheme. These double peaks likely correspond to the transitions between dry and wet seasons, which align well with the authors' best-fit findings.**

ATC: This is an insightful connection to make between these works. Some extended explanation added:

(ln 283) The double peak in the deposition seasonality of BF1 in the tropics (Figs. 8d, e) suggests that the fastest uptake may occur coincidentally with the ITCZ crossing the equator at the equinoxes (Fig. 8) rather than when the ITCZ is furthest from the equator, as in the prototype scheme. This implies that BF1 may reflect soil-moisture processes not captured in the prototype scheme. Paulot et al. (2024) have recently shown how a deposition scheme driven by three-hourly varying soil parameters from the Global Land Data Assimilation System (Rodell et al., 2004), and a low soil moisture activation threshold for bacterial uptake, produced a double-peak in $H_2$ in the tropics and better captured NH $H_2$ seasonality. This is distinct from their base simulation driven by monthly soil moisture, where NH sub-tropical $H_2$ peaked three months earlier than observations, comparable with the prototype scheme driven with monthly-mean ERA5 data in this work (Fig. 4b).

**Minor:**

**RC: Line 19: consider rephrasing as "...contributes to the formation of the greenhouse gas ozone...".**

ATC: Thank you for this suggestion, this is a clearer statement as the contribution to ozone formation involves a series of reactions. Becomes:

(ln 19) Hydrogen oxidation by OH additionally contributes to the formation of the greenhouse gases ozone and water, with the latter having a significant warming effect in the otherwise dry stratosphere (Sand et al., 2023; Warwick et al., 2023).

**RC: Lines 27–31: This doesn't read clearly at the moment. The long-term H$_2$ trend linked to CH$_4$ is clear; what remains less clear are shorter-term fluctuations, likely reflecting anthropogenic emissions and soil uptake increase/reduction.**

ATC: Thank you for pointing out this distinction between explanations for long-term trends and sub-decadal fluctuations. I think it is useful to refer to Derwent et al. (2023, Atmospheric Environment) and Paulot et al. (2024, ACP) here:

(ln 29) Multi-decadal increases in H_2 can mainly be attributed to increases in methane oxidation, while sub-decadal variations are more related to changes in the soil sink and H_2 emissions (Derwent et al., 2023; Paulot et al., 2024).

**RC: Line 68: clarify that "diffusion" refers to atmospheric (turbulent) dispersion, not soil diffusion.**

ATC: for clarity this is changed to:

(ln 67) … and atmospheric dispersion parameters tuned based on reproducing the observed SF_6 distribution.

**RC: Line 74: the estimate that only ~1% of bacterial biomass is hydrogen-oxidizing may be outdated. Greening and Grinter (2022, Nat. Rev. Microbiol) report up to 40% in forest soils and up to 90% in deserts.**

ATC: these new results are very important, in particular the recent findings of the ubiquity of the hydrogen oxidising community needs to be recognised clearly (Bay et al. 2021 and Greening and Grinter (2022) added as references). Becomes:

(ln 71) H_2 oxidising bacteria are ubiquitously distributed in soils (Schlegel, 1974; Khdhiri et al., 2015; Greening et al., 2016; Ji et al., 2017; Bay et al., 2021; Greening and Grinter, 2022).

**RC: Line 95: you may want to comment that you assume proportionality to h·f·g, but in the widely used formulation of Ehhalt and Rohrer (2013), deposition velocity scales as (h·f·g)^0.5.**

ATC: Thank you for pointing out that this is a useful note for the reader, their scaling with (fgh)^1/2 solution comes about for moist soils where the diffusive depth is limited. This paragraph becomes:

(ln 88) Analytically, the resultant H_2 uptake is the parallel sum of the potential flux limited by biological activity with the potential flux limited by diffusion (Bertagni et al., 2021). However, that formulation would require accurate quantification of each flux. Alternately, Ehhalt and Rohrer (2013) have formulated deposition velocity in moist soils to vary with (fgh)^1/2. However, in this work, the objective of the prototype scheme is to capture the seasonality in these processes while facilitating an analysis of how these processes drive the planetary H2 distribution. Therefore, we adopt an idealised formulation where the total uptake is proportional to the product of the normalised terms, fgh, and is scaled to achieve a total 57.2 Tg yr−1 average deposition (following Sand et al., 2023).

**RC: Figure 2: H$_2$ should not be italicized.**

ATC: fixed in figure.

**RC: Line 109: suggest rephrasing "carefully handled"**

ATC: rephrased for style improvement:

(ln 108) The effects of diffusive barriers are diagnosed by Bertagni et al. (2021). Their presence has the strongest limiting effect where the underlying soil diffusivity is highest but this is masked where the biological uptake is strongly limited due to lack of soil moisture, or by cold temperatures where there is snow cover.

**RC: Table 2: h, f, and g are dominant not only in Bertagni's formulation but also in that of Ehhalt and Rohrer.**

ATC: E+H (2013) now cited in Table 2.

**RC: Figure 8: the hatched regions in panel a are hard to distinguish when printed; also, I couldn't understand if the arrows should reach the "target" values, as these don't match those in panel c.**

ATC: hatches now produced in red (more clear on the blue-green-brown colouring) and figure caption re-written for clarity:

Figure 8. (a-c) Colours show $R_{BF}$ (Equation D1), which measures the performance of adjusted versions of the prototype deposition scheme at reproducing the best-fit deposition scheme, BF1. $R_{BF} = 1$ when the adjusted scheme performs as well as annual mean deposition with no seasonality, and $R_{BF} = 0$ when the deposition scheme reproduces BF1. In (a), only the amplitude of the deposition seasonality is adjusted, by scaling with a multiplier, $\alpha$. In (b), only the timing of the deposition seasonality is adjusted, by including an offset, $\Delta t$. Arrows indicate optimal adjustments at the latitudes of peaks in deposition seasonality in the prototype scheme (annotated A: 18∘S, B: 13∘N and C: 52∘N in each panel, see Fig. 7a) when $\alpha$ and $\Delta t$ are adjusted individually. (c) the optimal $R_{BF}$ achieved at latitudes A, B and C when $\alpha$ and $\Delta t$ are adjusted jointly. Seasonality of deposition for the prototype and BF1 schemes, integrated across three latitude bands: (d) 0-30°S; (e) 0-30°N; and (f) 30-60°N.

[Figure]

**Figure 8.** (a-c) Colours show $R_{BF}$ (Equation D1), which measures the performance of adjusted versions of the prototype deposition scheme at reproducing the best-fit deposition scheme, BF1. $R_{BF} = 1$ when the adjusted scheme performs as well as annual mean deposition with no seasonality, and $R_{BF} = 0$ when the deposition scheme reproduces BF1. In (a), only the amplitude of the deposition seasonality is adjusted, by scaling with a multiplier, $\alpha$. In (b), only the timing of the deposition seasonality is adjusted, by including an offset, $\Delta t$. Arrows indicate optimal adjustments at the latitudes of peaks in deposition seasonality in the prototype scheme (annotated A:18°S, B:13°N and C:52°N in each panel, see Fig. 7a) when $\alpha$ and $\Delta t$ are adjusted individually. (c) the optimal RBF achieved at latitudes A, B and C when $\alpha$ and $\Delta t$ are adjusted jointly. Seasonality of deposition for the prototype and BF1 schemes, integrated across three latitude bands: (d) 0-30°S; (e) 0-30°N; and (f) 30-60°N. ~~The ratio $R_{BF}$ (Equation D1) measuring how well adjusted versions of the prototype deposition scheme perform at reproducing the best-fit deposition scheme, BF1. (a) the prototype deposition seasonality is scaled by a multiplier, $\alpha$; (b) the deposition seasonality is offset in time by $\Delta t$; and (c) minimum $R_{BF}$ achieved when $\alpha$ and $\Delta t$ are varied at the latitudes of peaks in deposition seasonality in the prototype scheme (annotated A:18°S, B:13°N and C:52°N in each panel, see Fig. 7a). Arrows indicate adjustments that improve the performance at latitudes A,B and C, hatches indicate where the adjusted deposition schemes perform worse than the prototype scheme.~~

**RC: Figure 8e: I like this panel, showing a clear double peak instead of the expected single maximum, consistent with the comment above.**

ATC: this shows that the best-fit in broad bands that are unconstrained by the theoretical seasonality reproduce a deposition seasonality reflecting the dry-wet tropical seasons. Please see main response above.

**RC: Lines 389–390: I strongly support the authors' call for more hydrogen observations in tropical environments (atmosphere and soil both).**

ATC: I am optimistically looking forward to efforts from the observation community to achieve this.

---

## Author Response (AR3)

Dear Editorial Team,

Please find the latest changes:

Ln11: Intertropical Convergence Zone not abbreviated.

Ln13: 'TEXT' place holder removed

Ln390: 'Code and data availability. The 2D model code and output data are made available at https://doi.org/10.5281/zenodo.15082493.'

Many thanks,

Alex Tardito Chaudhri, David Stevenson